# A conserved YAP/Notch/REST network controls the neuroendocrine cell fate in the lungs

Yan Ting Shue[1,2], Alexandros P. Drainas[1,2], Nancy Yanzhe Li [3], Sarah M. Pearsall [4], Derrick Morgan[4], Nasa Sinnott-Armstrong [2], Susan Q. Hipkins[1,2], Garry L. Coles[1,2], Jing Shan Lim[1,2], Anthony E. Oro [3], Kathryn L. Simpson[4], Caroline Dive [4] & Julien Sage [1,2✉]

The Notch pathway is a conserved cell-cell communication pathway that controls cell fate decisions. Here we sought to determine how Notch pathway activation inhibits the neuroendocrine cell fate in the lungs, an archetypal process for cell fate decisions orchestrated by Notch signaling that has remained poorly understood at the molecular level. Using intratumoral heterogeneity in small-cell lung cancer as a tractable model system, we uncovered a role for the transcriptional regulators REST and YAP as promoters of the neuroendocrine to non-neuroendocrine transition. We further identified the specific neuroendocrine gene programs repressed by REST downstream of Notch in this process. Importantly, we validated the importance of REST and YAP in neuroendocrine to non-neuroendocrine cell fate switches in both developmental and tissue repair processes in the lungs. Altogether, these experiments identify conserved roles for REST and YAP in Notch-driven inhibition of the neuroendocrine cell fate in embryonic lungs, adult lungs, and lung cancer.

[1] Departments of Pediatrics, Stanford University, Stanford, CA, USA. [2] Departments of Genetics, Stanford University, Stanford, CA, USA. [3] Departments of Program in Epithelial Biology, Stanford University, Stanford, CA, USA. [4] Cancer Research UK Manchester Institute Cancer Biomarker Centre, University of Manchester, Manchester, UK. ✉email: julsage@stanford.edu

Cell fate decisions are critical to ensure proper organ development during embryogenesis and homeostasis in adult organisms. Accordingly, defects in signaling pathways regulating cell fate are associated with developmental disorders and diseases such as cancer. The Notch signaling pathway serves as a paradigm in studies of cell fate control (reviewed in refs. [1–5]). Schematically, in this cell–cell communication pathway, ligand molecules expressed on the surface of cells activate receptors on neighboring cells, resulting in receptor cleavage and the nuclear translocation of the NOTCH intra-cellular domain (NICD), which acts as a transcriptional regulator with its DNA binding partner RBP-J. NICD targets include regulators of the cell cycle, cell survival, and cell differentiation (reviewed in refs. [6,7]). Despite a large number of studies, however, the mechanisms by which Notch signaling is regulated and how it controls cell fate changes are still only partly understood.

The control of the neuroendocrine (NE) cell fate in the lung epithelium has provided a model to investigate Notch signaling in vivo (reviewed in refs. [8–11]). During embryonic development, ligand-expressing epithelial cells accumulate expression of ASCL1 (Achaete-Scute Family BHLH Transcription Factor 1) while activation of NOTCH receptors and their transcriptional target HES1 (Hes Family BHLH Transcription Factor 1) in neighboring cells results in ASCL1 downregulation and inhibition[12–14]. ASCL1 is a key activator of NE gene programs, including genes such as *Calca* coding for the calcitonin gene-related peptide CGRP[15–17]. Thus, ligand-high Notch-inactive cells express ASCL1 and give rise to NE cells while receptor-high Notch-active cells lose ASCL1 expression and give rise to non-NE lung epithelial cells[15,18–21]. A similar process is at play in the adult lung following injury where Notch activation in a subset of NE cells promotes the generation of non-NE epithelial cells such as club cells[22–24].

Remarkably, accumulating evidence indicates that developmental processes involving Notch signaling and decisions between NE and non-NE cell fates are also at play in small-cell lung cancer (SCLC), an aggressive and early disseminating form of lung cancer characterized by resistance to multiple therapeutic approaches and dismal survival rates (reviewed in ref. [25]). In mouse models, ASCL1 is critical for the initiation of SCLC[16,26]. SCLC cells can switch from a NE state to non/less-neuroendocrine (non-NE) states in response to various signals, and these non-NE states can affect the metastatic ability of SCLC cells as well as their response to chemotherapy[27–30]. In particular, mouse and human SCLC cells can acquire non-NE features upon activation of Notch signaling[29]. Notch-active non-NE SCLC cells lose the expression of classical NE markers, and express high levels of NICD targets such as HES1 and REST (RE1-silencing transcription factor, also known as neuron-restrictive silencer factor, NRSF). These HES1[high] non-NE SCLC cells are inherently more resistant to chemotherapy and also support the growth of NE SCLC cells in the tumor microenvironment[29]. Other transcription regulators associated with loss of classical NE features in SCLC include YAP, POU2F3, ATOH1, and MYC[31–37]. But how these factors control NE and non-NE cell states and how they possibly connect with Notch signaling remains poorly understood in SCLC and during development.

Here we seek to further investigate conserved molecular networks around Notch implicated in the transition from NE to non-NE cell states. We identify REST and YAP as key transcriptional regulators of this transition in multiple contexts. In SCLC, REST represses a set of neuroendocrine genes largely distinct from ASCL1 targets and is required for the complete repression of the NE program when SCLC cells transdifferentiate into a non-NE club cell-like fate. In non-cancer contexts, deletion of *Rest* results in a higher number of PNECs specified during early lung development and a delay in NE to non-NE transition

following repair after lung injury with naphthalene. Similarly, YAP is also found to be antagonistic to the neuroendocrine fate. When YAP is deleted following lung injury, PNECs are unable to transdifferentiate to the non-NE fate to replenish the epithelium of the airways. These findings provide further insights into the molecular mechanisms by which Notch signaling controls cell fate transitions in vivo.

## Results

**HES1-high non-NE SCLC cells have features of club cells**. To better characterize Notch-active HES1[high] non-NE SCLC cells, we sought to compare their transcriptional profile to that of HES1[low] NE SCLC cells. To this end, we conducted RNA-seq on cancer cell populations isolated by flow cytometry from *Rb/p53/p130* triple knockout (*TKO*) tumors growing in the lungs of *Rb^{fl/fl};p53^{fl/fl}; p130^{fl/fl};Hes1^{GFP/+}* mice following intra-tracheal instillation of an adenovirus expressing Cre (Ad-CMV-Cre). In these *TKO;Hes1^{GFP/+}* tumors, a green fluorescent protein (GFP) reporter is expressed from the regulatory regions of the endogenous *Hes1* gene (Fig. 1a and Supplementary Fig. 1a). The transcriptomic analysis showed clear separation between GFP[high] and GFP[neg] SCLC cells by principal component analysis (PCA) (Supplementary Fig. 1b). We identified 2518 upregulated and 2017 downregulated genes with corrected *p*-value < 0.05 and log2-fold change of >1.0 and <−1.0 (Supplementary Fig. 1c and Supplementary Data 1). Gene set enrichment analysis (GSEA) confirmed Notch pathway enrichment in the GFP[high] cells (Supplementary Fig. 1d). Gene ontology (GO) term analysis of the differentially expressed genes showed downregulation of NE differentiation in GFP[high] cells, with upregulation of programs involved in cell adhesion, collagen-containing extracellular matrix, blood vessel morphogenesis and inflammation (Supplementary Fig. 1e, f and Supplementary Data 2). GFP[high] cells also expressed lower levels of genes typically associated with pulmonary neuroendocrine cells (PNECs) and higher levels of markers commonly expressed in non-NE epithelial cells such as ciliated cells, club cells, and alveolar type 1 (AT1) and type 2 (AT2) cells compared to GFP[neg] NE SCLC cells (Fig. 1b).

The expression of a variety of non-NE differentiation markers in GFP[high] cells from *TKO;Hes1^{GFP/+}* tumors raised the possibility that this cell population was heterogeneous. However, using single-cell RT-qPCR, we found that GFP[high] SCLC cells had a relatively homogenous pattern of expression for the genes tested, including high levels of genes coding for club cell markers as well as lower levels of AT1 cell markers, which are also expressed at low levels in normal club cells[24] (Fig. 1c and Supplementary Fig. 1g). The fold-change differences between the bulk RNA-seq and single-cell RT-qPCR data are likely due to differences in absolute transcript levels and the different platforms (Supplementary Data 1). Immunostaining only detected expression of the club cell marker CC10 (also known as SCGB1A1) and not the AT1 marker AGER on tumor sections (Fig. 1d). CC10 also frequently co-localized with HES1, indicating that HES1[high] non-NE SCLC cells do not express detectable levels of this AT1 marker (Fig. 1e, f).

To investigate the role of Notch signaling in the generation of these non-NE club cell-like SCLC cells, we deleted *Rbpj* in *TKO* mice at the time of tumor initiation. Loss of RBP-J did not affect overall tumor burden but led to a significant reduction in HES1[+] and CC10[+] cells in the tumors (Fig. 1g–i and Supplementary Fig. 2a–c). The remaining few CC10[+] cells were found to express detectable levels of RBP-J and hence, likely to be infiltrating normal club cells (Supplementary Fig. 2d). A similar requirement for RBP-J/Notch signaling in the NE to non-NE transdifferentiation process was recently described in a different mouse model of SCLC[38].

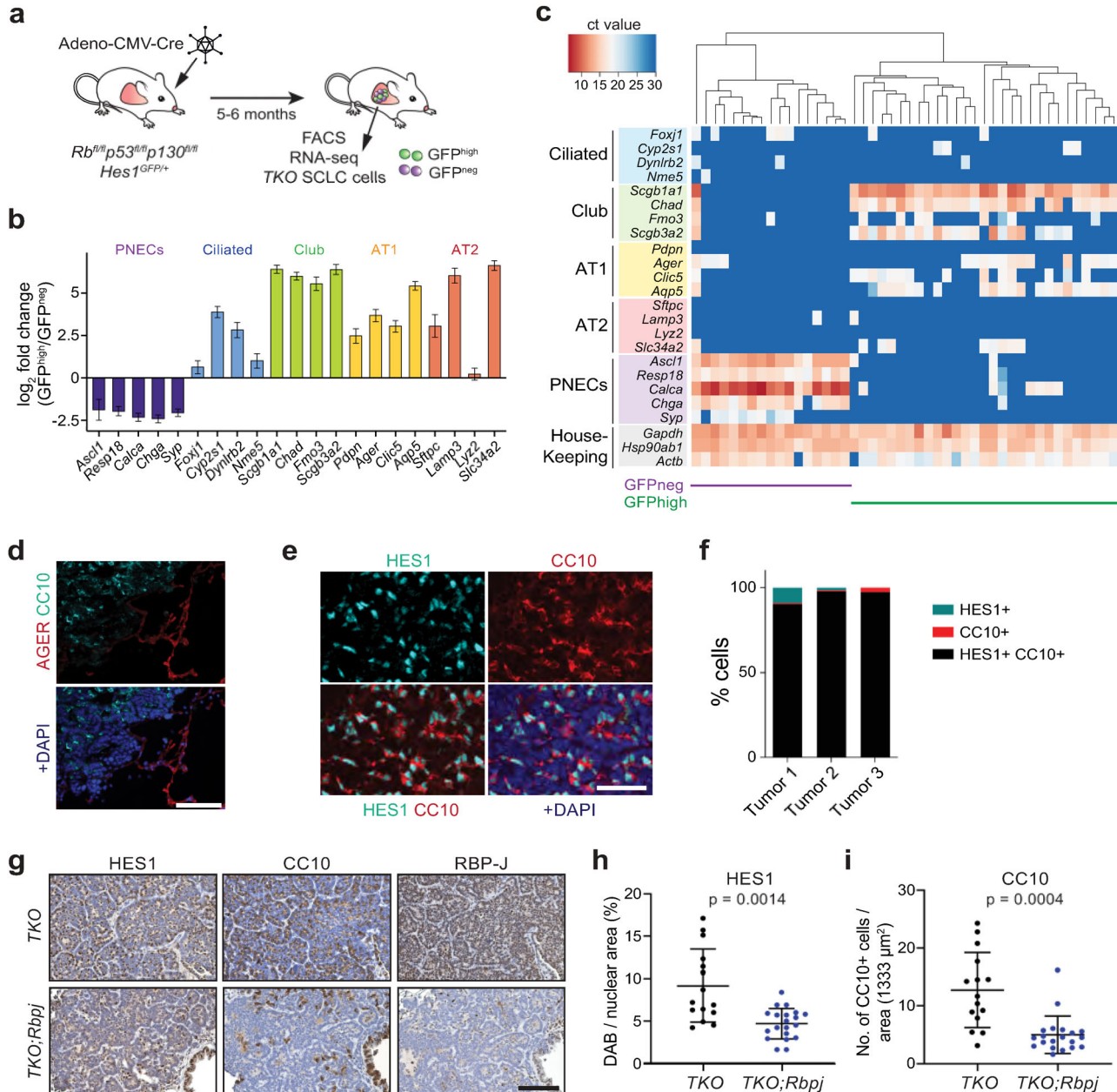

**Fig. 1 HES1-positive non-neuroendocrine cells in mouse SCLC tumors have features of club epithelial cells. a** Schematic representation of the $Rb^{fl/fl};p53^{fl/fl};p130^{fl/fl};Hes1^{GFP/+}$ mouse model of SCLC ($TKO;Hes1^{GFP/+}$) in which HES1+ non-neuroendocrine SCLC cells generated by Notch pathway activation can be visualized and isolated by GFP$^{high}$ expression from the $Hes1$ locus, along with GFP-negative (GFP$^{neg}$) neuroendocrine SCLC cells. **b** Expression of classical markers of the indicated lung epithelial cell types from RNA-seq data in GFP$^{high}$ cells compared to GFP$^{neg}$ cells ($n = 4$). Error bars, log2 fold change estimate ± SE. **c** Single-cell RT-qPCR analysis (Fluidigm) of GFP$^{high}$ and GFP$^{neg}$ SCLC cells ($n = 46$) isolated from $TKO;Hes1^{GFP/+}$ tumors for markers of lung epithelial cells. See Supplementary Fig. 1g for an independent experiment. In this sensitive assay, all GFP$^{high}$ cells express $Scgb1a1$, which codes for the club cell marker CC10. **d** Representative immunofluorescence images for AGER (red) and CC10 (cyan, encoded by $Scgb1a1$) on a mouse SCLC tumor section ($n = 1$ experiment, multiple tumors were stained). DAPI stains the DNA in blue. Scale bar, 100 µm. **e** Representative immunofluorescence images for HES1 (cyan) and CC10 (red) on a mouse SCLC tumor section ($n = 2$ independent experiments). DAPI stains the DNA in blue. Scale bar, 50 µm. **f** Quantification (%) of cells that are either HES1+, CC10+, or HES1+ CC10+ in $n = 3$ mouse tumors as in (**e**). **g** Representative immunohistochemistry images of HES1, CC10, and RBP-J staining (brown signal) in $TKO$ and $TKO;Rbpj^{fl/fl}$ tumors (hematoxylin counterstain). Scale bar, 100 µm. **h, i** Quantification of HES1 staining (**h**) and CC10 staining (**i**) in tumor sections from (**g**) ($n = 15$ tumors from 3 mice for $TKO$ and 20 tumors from 4 mice for $TKO;Rbpj^{fl/fl}$). Unpaired $t$-test with Welch's correction, data represented as mean ± s.d. See also Supplementary Figs. 1 and 2 and Supplementary Data 1 and 2. Source data are provided as a Source Data file.

Therefore, GFP$^{high}$ SCLC cells in $TKO;Hes1^{GFP/+}$ tumors are a relatively homogeneous population of club cell-like cancer cells generated by activation of Notch signaling. These club cell-like phenotypes in SCLC tumors are reminiscent of both the transdifferentiation of wild-type NE epithelial cells into club cells in the adult lungs in response to injury via activation of Notch signaling[23,24] and the generation of NE and non-NE cell during embryonic lung development[11,19]. The co-optation of these developmental and regeneration pathways in cancer suggested that SCLC may provide a useful model to further investigate the

mechanisms of cell fate switch upon activation of Notch signaling in NE cells.

**REST is a regulator of the NE to non-NE transition in SCLC.** We previously identified REST as a candidate effector of Notch signaling in SCLC cells from microarray studies[29]. From our RNA-seq dataset and single-cell RT-qPCR analysis, *Rest* was significantly upregulated in HES1$^{high}$/GFP$^{high}$ non-NE SCLC cells (Supplementary Figs. 1c, 3a and Supplementary Data 1) and was again identified as the top candidate transcription factor repressing the downregulated genes during the cell fate transition (Supplementary Fig. 3b and Supplementary Data 1). In our previous study, REST overexpression in NE SCLC cells was sufficient to decrease the mRNA levels of the NE genes *Chga*, *Chgb* and *Syp*, but not of *Ascl1* and the ASCL1 target *Calca* (coding for CGRP)[29]. ASCL1 downregulation and inhibition is viewed as a key event downstream of Notch in the inhibition of the NE cell fate[15–17]. Hence, we sought to determine the extent of overlap between REST and ASCL1 targets in SCLC.

ChIP-seq analysis of REST in GFP$^{high}$ SCLC cell lines in culture identified 1317 gene targets (FDR cutoff < 0.05 and more than 2-fold change) (Fig. 2a and Supplementary Data 3). About half of the binding sites were within 3 kb of transcription start sites and HOMER motif analysis identified REST binding motif as the top hit (Supplementary Fig. 3c, d). Upon ectopic expression of REST in NE SCLC cells in culture, 99 and 205 genes were downregulated by RNA-seq after 48 h and 5 days, respectively, with corrected *p*-value < 0.05 and log2 fold change of <−1.0 (Fig. 2a, Supplementary Fig. 3e, f and Supplementary Data 4). Combining the ChIP-seq and the 5-day REST overexpression RNA-seq datasets, we identified 141 high-confidence REST targets in NE SCLC cells (Fig. 2a and Supplementary Data 5). ASCL1 targets were obtained from a ChIP-seq dataset generated from primary *TKO* tumors[16], identifying 3008 candidate gene targets following our analysis pipeline (Fig. 2a and Supplementary Data 6). Comparing the genomic regions bound by ASCL1 and REST in the ChIP-seq datasets showed little overlap in their binding sites (Supplementary Fig. 3g). To generate high-confidence ASCL1 targets, we knocked down *Ascl1* in NE SCLC cells for RNA-seq analysis after 3 days. In this context, ASCL1 knock-down has detrimental effects on population growth (Supplementary Fig. 4a, b). 180 and 493 genes were downregulated by sh*Ascl1* #1 and sh*Ascl1* #2, respectively, with corrected *p*-value < 0.05 and log2 fold change of <−1.0 (Supplementary Fig. 4c–e and Supplementary Data 7). Filtering the downregulated genes with no fold change cut-off and comparing with genes identified in the ASCL1 ChIP-seq identified 250 genes (Supplementary Fig. 4f and Supplementary Data 7). GO terms analysis showed enrichment in Notch signaling and nervous system development pathways (Supplementary Fig. 4g and Supplementary Data 8). Out of these, only 9 overlapped with the 141 high-confidence REST targets (Supplementary Fig. 4h). As the ASCL1 knockdown was detrimental to the cells, it was possible that genes with more subtle changes would be overlooked using this approach. Thus, we decided to instead narrow down the list of genes from the ASCL1 ChIP-seq to 559 by filtering out those that were not downregulated in GFP$^{high}$ cells (in which ASCL1 is expressed at very low levels) (Fig. 2a and Supplementary Data 5). By doing so, the overlap between ASCL1 targets and REST targets in SCLC cells was still only 43 genes (Fig. 2a and Supplementary Data 5). Examples of NE genes specifically induced by ASCL1 include *Calca* and *Dll3*, while genes specifically repressed by REST include *Chga*, *Chgb* and *Resp18*, and common genes include *Kcnc1* and *Olfm1* (Fig. 2b, c). Notably, GO terms analysis for both lists of targets

were similar but not identical (Fig. 2d and Supplementary Data 9). Even in the same categories, many genes were regulated by only one of the two transcriptional regulators (Fig. 2e).

Together, these analyses show that REST targets are largely distinct from ASCL1 targets and support a model in which both inhibition of the pro-NE transcription factor ASCL1 and activation of the REST repressor contribute to the NE to non-NE cell fate change observed in SCLC downstream of Notch signaling activation. These results led us to test the functional role of REST in Notch-driven NE to non-NE transitions in vivo.

**REST is required for complete repression of SCLC NE programs.** To first investigate a possible requirement for REST in the NE to non-NE transition in vivo in SCLC, we crossed *p53*$^{fl/fl}$*Rb*$^{fl/fl}$*p130*$^{fl/fl}$ mice to a conditional allele of *Rest* (*Rest*$^{fl}$)[39] and analyzed tumors in triple and quadruple mutant mice upon Ad-CMV-Cre delivery to the lungs of these mice. In this model, the vast majority of tumors are initiated in non-NE lung epithelial cells[40]. *Rest* deletion was efficient in this context, as assessed by in situ hybridization on lung sections (Supplementary Fig. 5a, b). We observed a reduction in tumor numbers in *TKO;Rest* quadruple mutant mice compared to *TKO* controls while *TKO;Rest* tumors trended to be slightly larger than *TKO* tumors (Fig. 3a–c). Expression of the individual markers HES1 and CC10 was similar on sections from *TKO* and *TKO;Rest* tumors (Supplementary Fig. 5c–e), suggesting that loss of REST is not sufficient to completely block the induction of these two non-NE markers. To gain a broader and unbiased view of the phenotypes related to REST loss, we isolated NE and non-NE cells from *TKO* and *TKO;Rest*$^{fl/fl}$ mutant mice (which were not crossed to the *Hes1*$^{GFP}$ allele) using the cell surface markers NCAM1 and ICAM1 based on the RNA-seq and FACS analysis of GFP$^{high}$ and GFP$^{neg}$ SCLC cells (Supplementary Data 1, Supplementary Fig. 6a,b and Fig. 3d). Cell populations isolated following this protocol grew in culture similar to NE and non-NE cells isolated based on HES1 levels[29], with NCAM1$^{high}$ ICAM1$^{low}$ NE cells classically forming loosely adherent clusters and NCAM1$^{low}$ ICAM1$^{high}$ non-NE cells growing in 2D layers on the plate (Fig. 3e). These two cell populations clustered separately by RNA-seq analysis and in a manner similar to the *Hes1*$^{low}$/GFP$^{neg}$ and *Hes1*$^{high}$/GFP$^{high}$ SCLC cell lines, respectively, with complete deletion of *p53*, *Rb* and *p130* alleles confirmed by PCR on genomic DNA, further validating the isolation protocol (Supplementary Fig. 6c–e). Importantly, we observed separate clustering of the NCAM1$^{low}$ ICAM1$^{high}$ non-NE samples in the PCA analysis when comparing the *TKO* and *TKO;Rest* mutant cell lines (Fig. 3f). 240 genes were differentially expressed between the two groups, with the majority being upregulated (217 genes) with the loss of REST, in concordance with its role as a transcription repressor (Fig. 3g and Supplementary Data 10). Notably, the genes not repressed in *Rest* mutant cells were significantly enriched in NE genes with 76 of them also identified as REST targets in our ChIP-seq/RNA-seq analysis (Fig. 3h, i and Supplementary Data 10). Of the 141 genes which did not overlap with our list of REST targets (possibly because cell lines and primary tumors may not be exactly the same), transcription factor analysis showed REST to be the top candidate transcriptional regulator binding in the vicinity of these genes, suggesting that a number of these genes are REST targets (Fig. 3i and Supplementary Data 11). Some of the genes (e.g., *L1cam*) were identified in the REST ChIP-seq experiment but did not meet the threshold for downregulation in the REST overexpression RNA-seq data. *Ascl1*, which is not a REST target, showed no differential expression in REST wild-type or mutant non-NE cells in contrast to the REST targets *Chga* and *Syp* (Supplementary Fig. 6f).

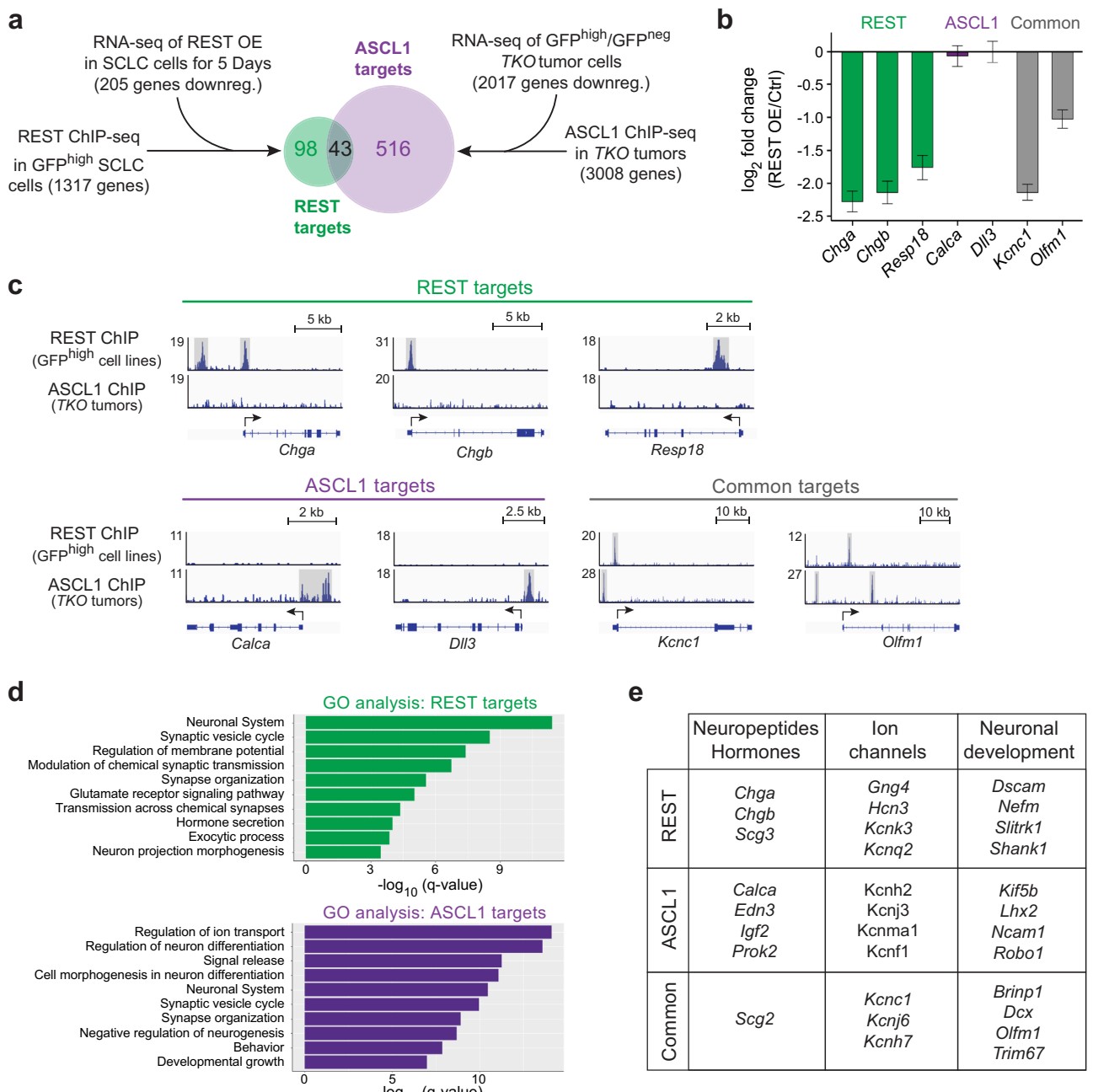

**Fig. 2 REST and ASCL1 control largely non-overlapping programs in SCLC cells. a** Experimental workflow to identify REST and ASCL1 candidate target genes. GFP^high cells are non-NE SCLC cells from *TKO;Hes1^GFP/+* tumors while GFP^neg cells represent NE SCLC cells. The mouse SCLC cell line used for the RNA-seq of REST overexpression (OE) is the neuroendocrine KP1 cell line. **b** Expression of selected neuroendocrine and neuronal-related genes from RNA-seq data of 5-day REST overexpression (*n* = 3). Error bars, log2 fold change estimate ± SE. **c** ChIP-seq data showing REST and ASCL1 binding near/within their respective target genes in (**b**). **d** Gene ontology (GO) analysis of candidate REST and ASCL1 targets. Note the similar GO terms between the two factors. **e** Examples of REST, ASCL1 and shared target genes in different gene categories. See also Supplementary Figs. 3 and 4 and Supplementary Data 3–9.

Together, these data indicate that REST is the major regulator of the derepressed genes when *Rest* is deleted in non-NE SCLC cells. These experiments also conclusively identify REST as a key regulator of the Notch-driven NE to non-NE transition in SCLC in vivo.

**Loss of REST delays the NE to non-NE transition after injury.** Treatment of mice with naphthalene induces cell death in most club cells, and part of the tissue repair following this injury is due

to PNECs that proliferate and differentiate into club cells in response to Notch pathway activation[24,41–43]. Because of the similarities between SCLC tumors and lung repair after injury regarding the Notch-driven NE to non-NE transition, we next sought to examine a possible requirement for REST in this other in vivo context.

First, to determine *Rest* expression in the adult mouse lung, we performed single-cell RT-qPCR analysis of lung epithelial cells dissociated from the lungs of adult *Chga-GFP* mice, which have constitutive GFP expression driven by the promoter of the NE

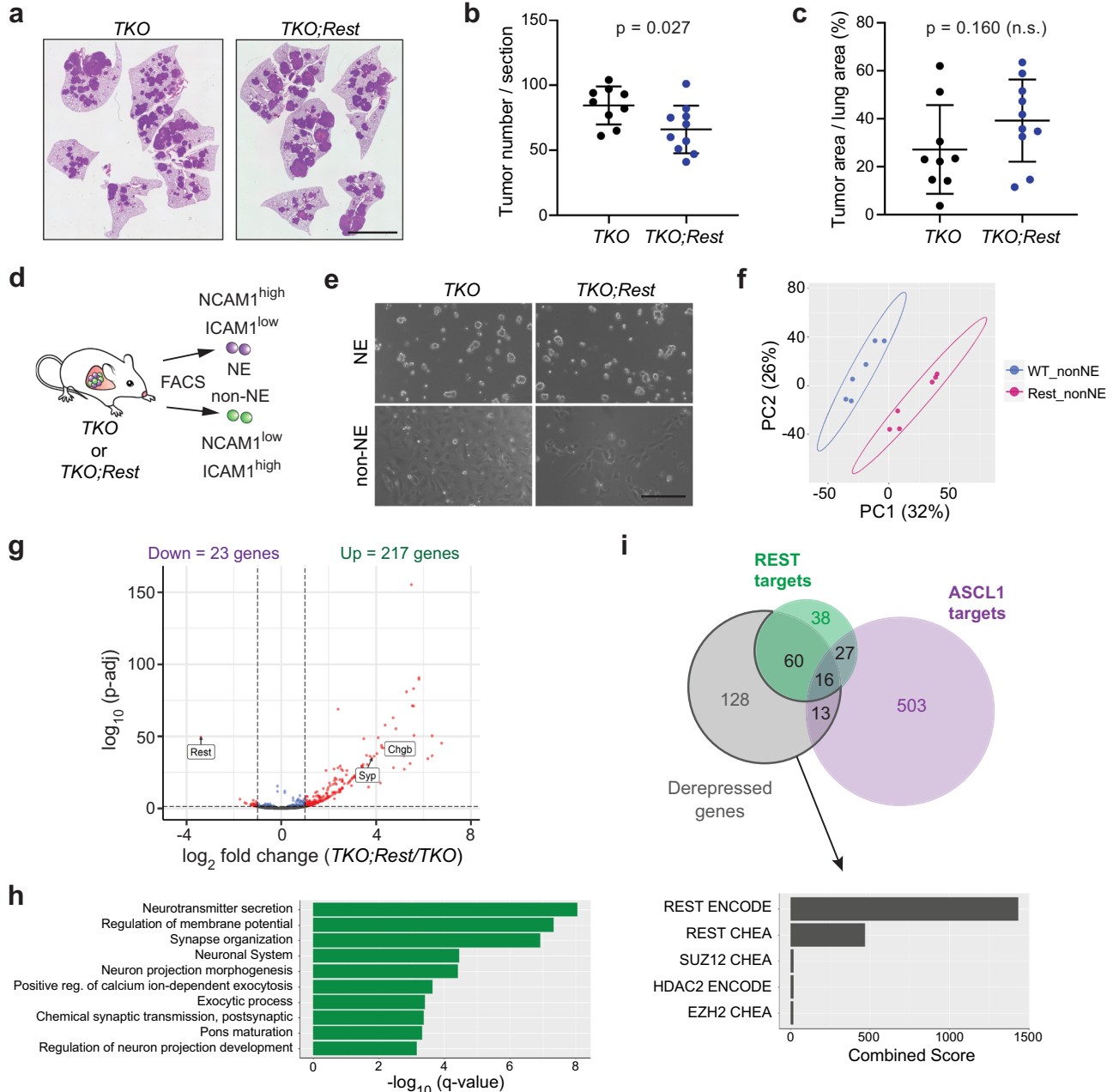

**Fig. 3 Loss of REST prevents the repression of its neuroendocrine targets in SCLC cells. a–c** Representative images of lung sections from *TKO* and *TKO;Rest^{fl/fl}* mice 5 months after Ad-CMV-Cre (*TKO;Rest*, quadruple mutant), stained with hematoxylin and eosin (H&E). Scale bar, 5 mm. Quantification of tumor numbers (**b**) and tumor area (**c**) ($n = 9$ mice for *TKO* and 10 mice for *TKO;Rest^{fl/fl}*). Error bar, mean ± s.d. **d** and **e** Schematic representation (**d**) and representative brightfield images (**e**) of neuroendocrine (NE) and non-NE cells sorted from *TKO* and *TKO;Rest* mutant tumors based on NCAM1 and ICAM1 expression. Scale bar, 200 μm. **f** Principal component analysis (PCA) of RNA-seq data comparing non-NE cell lines generated from *TKO* and *TKO;Rest* mutant tumors ($n = 6$ per genotype). WT wild-type for *Rest*; "Rest", knockout for *Rest*. **g** Volcano plot of differentially expressed genes from RNA-seq data of *TKO* and *TKO;Rest* mutant non-NE cell lines. *Rest* and the classical neuroendocrine markers *Syp/Chgb* are highlighted (significant genes with more than 2-fold change and *p*-adj value < 0.05 are in red). Wald test with Benjamini–Hochberg correction. **h** Gene Ontology (GO) analysis of the upregulated genes in the *TKO;Rest* mutant non-NE cell lines. **i** Venn diagram showing the amount of overlap between derepressed genes of *TKO;Rest* mutant non-NE cell lines with REST and ASCL1 targets identified in Fig. 2. Bar graph showing top 5 candidate transcription factors from Enrichr analysis (ENCODE and ChEA Consensus TFs from ChIP-X) on the 141 derepressed genes that did not overlap with our REST targets. See also Supplementary Figs. 5 and 6 and Supplementary Data 10 and 11. Source data are provided as a Source Data file.

gene and REST target *Chga*[44] (Supplementary Fig. 7a). As expected, based on the role of REST as a suppressor of the NE fate in SCLC, *Rest* expression was undetectable in GFP-expressing PNECs from *Chga-GFP* mice but present in other non-NE lung epithelial cell types (Fig. 4a). The analysis of a distinct single-cell

RNA-seq dataset of mouse lung epithelial cells[24] showed a similar result (Supplementary Fig. 7b).

To compare lung repair in wild-type and *Rest* mutant mice, we crossed *Rest^{fl/fl}* mice to *Ascl1^{CreER/+};Rosa26^{LSL-Tom/LSL-Tom}* mice. In these mice, PNECs were first labeled with Tomato by injecting

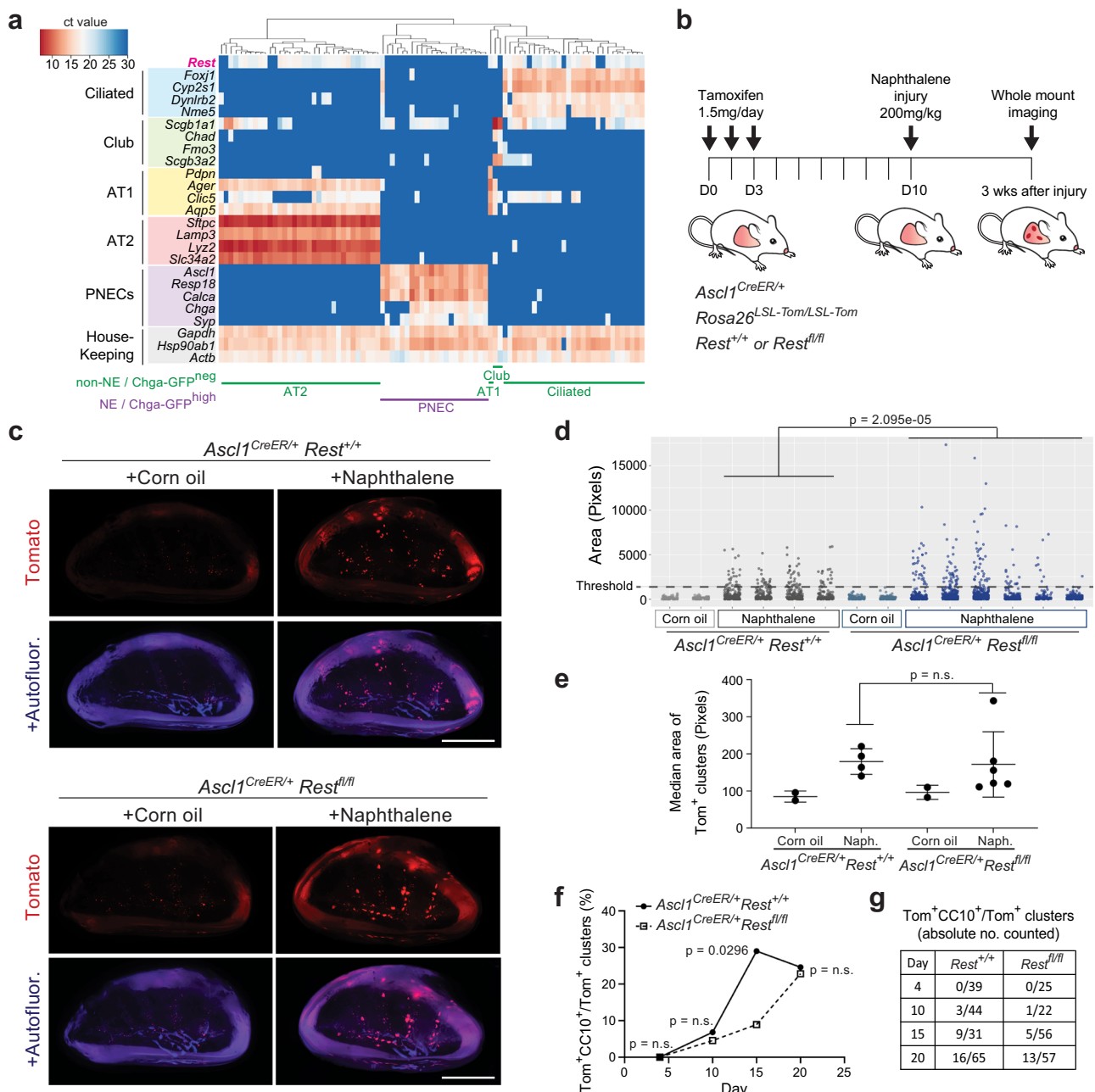

**Fig. 4 Loss of REST delays pulmonary neuroendocrine cell fate transition in response to naphthalene injury. a** Single-cell RT-qPCR analysis of *Rest* expression in pulmonary neuroendocrine cells (PNECs) and non-NE lung epithelial cells from normal lungs of two *Chga-GFP* adult mice (n = 87 cells). **b** Timeline of the PNEC lineage tracing and naphthalene-induced lung injury experiment. Tom, Tomato fluorescent reporter. **c** Whole-mount fluorescence images of left lung lobes 3 weeks after injury. Images representative of n = 2 independent experiments. Scale bar, 5 mm. **d, e** Quantification (**d**) of normal neuroepithelial bodies (NEBs) and PNEC outgrowths after injury as in (**c**) with each column representing a single mouse left lung lobe and each data point a tomato-positive cluster in that mouse. A Wilcoxon rank sum test, two-sided, was conducted on the combined PNEC outgrowths from all the mice in each respective group that are above the threshold size of the non-injured controls (n = 107 for *Rest*^+/+, n = 146 for *Rest*^fl/fl). Median area (**e**) of all Tom^+ clusters per mouse as in (**d**) (n = 2 mice for corn oil controls, 4 mice for naphthalene-treated *Ascl1*^CreER/+;*Rest*^+/+ and 6 mice for naphthalene-treated *Ascl1*^CreER/+;*Rest*^fl/fl). Unpaired *t*-test, data represented as mean ± s.d. **f, g** Percentage (**f**) and absolute number counted (**g**) of Tom^+CC10^+ clusters over total number of Tom^+ clusters at various timepoints after naphthalene injury (n = number of clusters as indicated in table (**g**) totaled from representative left lung lobe sections of 3–4 mice per timepoint). Fisher's exact test, two-tailed. See also Supplementary Fig. 7. Source data are provided as a Source Data file.

tamoxifen to activate the Cre recombinase expressed from the *Ascl1* locus. Mice carrying the floxed *Rest* alleles also had *Rest* deleted simultaneously without any observable effect on the PNEC since it is not expressed at this point (Supplementary Fig. 7c). This was followed by lung injury with naphthalene and a recovery period of three weeks (Fig. 4b). By quantifying outgrowths from labeled PNECs based on the size of Tomato-positive clusters, we found that loss of *Rest* resulted in the formation of much larger outgrowths with no change in the overall number or median area (Fig. 4c–e and Supplementary

Fig. 7d). PNECs were also able to transdifferentiate to a club cell fate marked by expression of CC10 as well as to downregulate the SYP protein (Supplementary Fig. 7e). We speculated that the increase in size of outgrowths might be due to a delay in terminal differentiation to club cells, allowing more time for transit amplifying cells to proliferate. To investigate this possibility, we conducted a time-course experiment quantifying the number of transdifferentiated Tomato-positive clusters across a period of 20 days, indeed, we observed the proportion of CC10$^+$ Tom$^+$ clusters peaking at Day 20, 5 days later than the control (Fig. 4f, g and Supplementary Fig. 7f). Hence, in this injury model, the loss of *Rest* does not increase the ability or probability of PNECs to transdifferentiate upon injury but results in a delay in differentiation to club cells, allowing more time for cells to proliferate during the transit amplification stage, explaining the larger size of the Tomato-labeled outgrowths.

These experiments demonstrate REST's involvement in Notch-controlled tissue repair in the adult lung epithelium.

**REST regulates PNEC specification during lung development.** Notch signaling also plays a central role in cell fate decisions between NE and non-NE cells during embryonic lung development[11,19]. To test the possibility of REST also playing a role in this setting, we crossed *Rest$^{fl/fl}$* mice to *Shh$^{Cre}$* mice, thereby deleting *Rest* in early lung progenitor cells and the entire lung epithelium. The analysis of lungs one day before birth (E18.5) showed a significant increase in the number of CGRP$^+$ PNECs and NEBs (neuroendocrine bodies) upon loss of REST (Fig. 5a–c). PNECs were also more often found in larger clusters in the mutant lungs as compared to controls (Fig. 5d, e). BrdU analysis showed that this increase was unlikely to be due to more proliferation (Supplementary Fig. 8a, b). RT-qPCR of entire lung lobes from *Rest* mutant mice found a 2-fold increase in non-*Rest* targets, *Ascl1* and *Calca*, in agreement with the doubled amount of PNECs. REST targets, *Chga*, *Chgb* and *Syp*, in contrast showed more than a 2-fold increase, with *Chga* being expressed a few hundred folds higher than control (Fig. 5f). In situ hybridization showed that *Chga* became expressed in non-NE lung epithelial cells, as confirmed by co-localization with E-cadherin, upon *Rest* deletion in embryonic lungs (Fig. 5g–i). Thus, in the absence of REST, non-NE epithelial cells of the developing lung misexpress REST-repressed NE genes.

While *Rest* is a direct target of NICD[29], regulatory networks often involve positive and negative feedback loops. We were particularly intrigued by RNA-seq and ChIP-seq data in SCLC cells showing regulation by REST of the expression of the *Dner* gene coding for an atypical and poorly studied NOTCH ligand (DNER, Delta/Notch like epidermal growth factor related receptor)[45–47] (Supplementary Fig. 8c, d), suggesting a possible regulation of Notch signaling by REST. We confirmed expression of DNER in *TKO* SCLC tumors and in PNECs (Supplementary Fig. 8e, f). RT-qPCR of E18.5 embryonic lungs also showed an increase in *Dner* expression in *Rest* knockouts compared to controls (Supplementary Fig. 8g). These observations and the existence of genetic variants in *DNER* associated with the development of diabetes, suggesting a link with neuroendocrine phenotypes[48], led us to investigate the role of DNER in the NE fate in mice. We generated *Dner* knockout mice by CRISPR/Cas9 targeting in mouse oocytes and confirmed deletion (Supplementary Fig. 8f). However, we did not observe any difference in PNEC numbers in *Dner* mutant E18.5 embryonic lungs (Supplementary Fig. 8h, i). We also found no effect of *Dner* inactivation in the context of injury in adult mice (Supplementary Fig. 8j, k). Thus, feedback control of Notch signaling by REST remains an open question, but the atypical NOTCH ligand and REST target DNER does not control the NE fate in the contexts examined.

**ATAC-seq identifies a TEAD signature in non-NE SCLC cells.** Our data demonstrate that REST is a bona fide downstream mediator of Notch signaling in NE lung cells, and that REST targets are distinct from ASCL1 targets. But these data did not exclude the possibility of other regulators being involved in NE to non-NE transitions. We examined differences in chromatin accessibility between the NE and non-NE SCLC cells using ATAC-seq[49] as a different unbiased way to investigate this cell fate transition (Fig. 6a). Principal component analysis (PCA) on the ATAC-seq data showed clear separation between NE and non-NE cells (Supplementary Fig. 9a). We found in total 80,367 differentially accessible peaks, with 39,539 more closed and 40,828 more opened in the non-NE cells as compared to the NE cells with corrected *p*-value < 0.001 and log2 fold change of <−1.0 or >1.0 (Fig. 6b). Motif analysis on the more accessible peaks ranked FOX family motifs and TEAD motifs at the top of the list of transcription factor motifs enriched in non-NE cells (Fig. 6c and Supplementary Data 12). However, as the FOX family motifs were also top hits in the analysis of the closed peaks (Supplementary Data 12), we decided to focus on the TEAD motifs. TEAD transcription factors are often (but not exclusively) associated with YAP1/TAZ factors downstream of the Hippo signaling pathway (reviewed in refs. [50,51]). As noted above, YAP1 has been associated with non-NE phenotypes in SCLC cells[32–34]. In agreement with the ATAC-seq findings, *Yap1* and *Wwtr1* (*Taz*) as well as their downstream targets were upregulated in GFP$^{high}$ cells in the RNA-seq dataset (Fig. 6d). Immunostaining confirmed that a large proportion of YAP1-positive cells also co-express the non-NE club cell marker CC10 in mouse SCLC tumor sections (Supplementary Fig. 9b, c).

These observations raised the possibility of YAP1 being capable of activating the non-NE program in SCLC cells. We over-expressed YAP1-GFP or GFP in NE mouse SCLC cells and found that YAP1-expressing cells became adherent, a feature of non-NE cells, after 14 days (Fig. 6e–g). NOTCH2 and HES1 were upregulated in YAP1-overexpressing cells while the NE markers ASCL1 and UCHL1 were downregulated (Fig. 6h). RNA-seq analysis 5 days after expression of YAP1 showed significant upregulation of *Notch2* and the Notch pathway targets *Hes1* and *Rest* (Fig. 6i and Supplementary Data 13). This observation was confirmed by RT-qPCR on independent samples (Supplementary Fig. 9d). GO term analysis of upregulated genes showed genes induced by YAP1 to be involved in blood vessel morphogenesis and cell adhesion, similar to *Hes1$^{high}$*/GFP$^{high}$ cells from *TKO;Hes1$^{GFP/+}$* mice (Supplementary Fig. 9e and Supplementary Data 13). YAP1 has been shown to directly upregulate *Notch2* in liver cells[52]. We conducted ChIP-qPCR on GFP$^{high}$ SCLC cell lines generated from *TKO;Hes1$^{GFP/+}$* SCLC tumors and found YAP1 binding to the promoter region of *Notch2* (Fig. 6j). Together, these observations suggest that increased activity of YAP1/TEAD can sensitize NE cells to Notch signaling by induction of NOTCH2 and is sufficient to promote the NE to non-NE cell fate switch.

With our data placing YAP1 upstream of Notch signaling, we expected to detect YAP1 expression when Notch signaling is blocked in tumor sections from *TKO;Rbpj* mutant mice. Instead, we found that YAP1 levels were reduced similarly to HES1 and CC10 upon loss of RBP-J (Supplementary Fig. 9f, g). Thus, it is possible that YAP1 expression is both upstream and downstream of Notch signaling, or that the two pathways enforce each other's activity in this context, as suggested by a recent study[38].

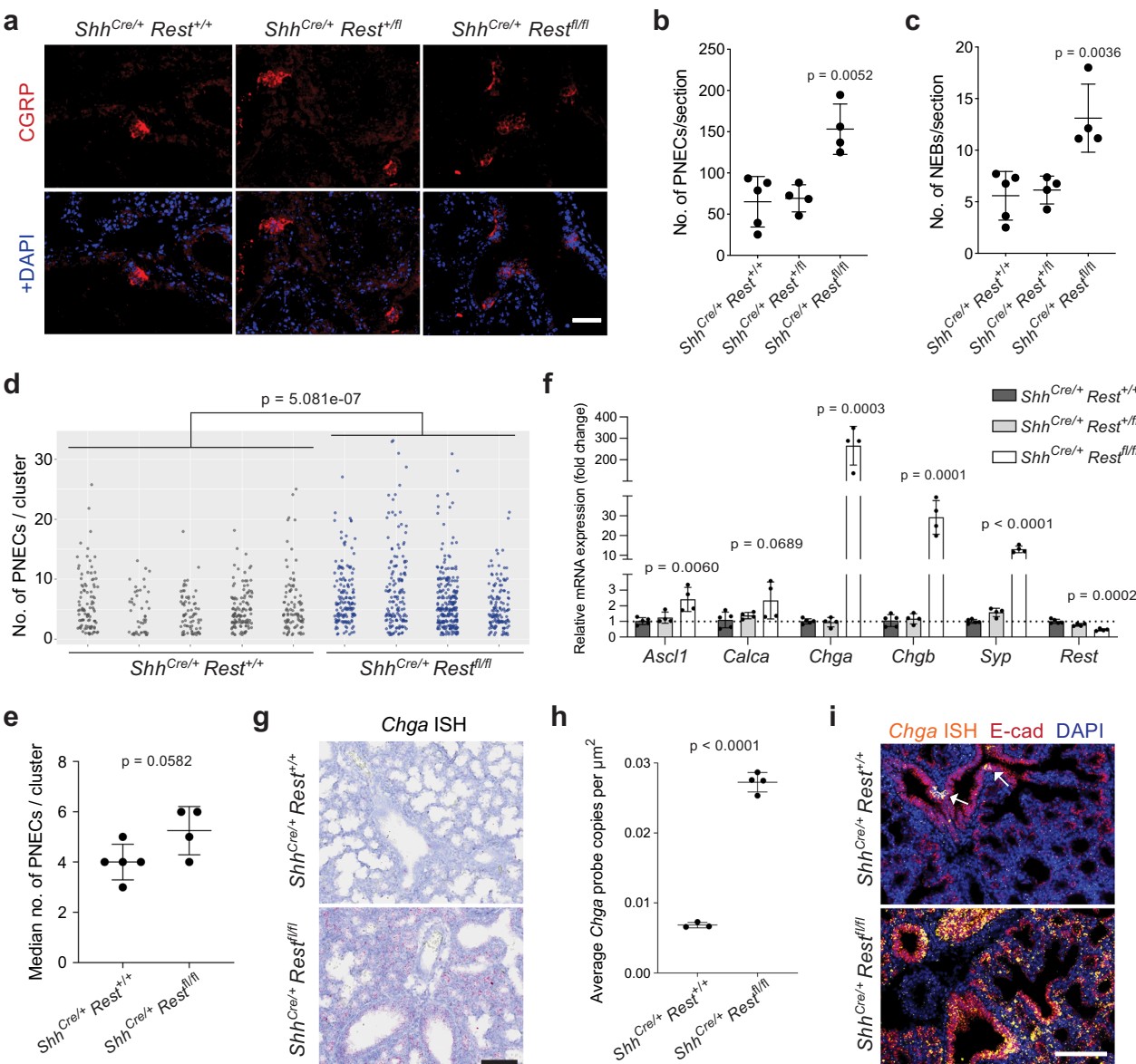

**Fig. 5 REST is antagonistic to pulmonary neuroendocrine cells specification in normal lung development. a** Immunostaining for CGRP, a marker of pulmonary neuroendocrine cells (PNECs) on sections from E18.5 $Shh^{cre/+}$;$Rest^{+/+}$, $Rest^{+/fl}$, and $Rest^{fl/fl}$ lungs. DAPI stains DNA in blue. Scale bar, 50 µm. **b–d** Quantification of the average number of CGRP$^+$ PNECs (**b**) and neuroepithelial bodies (NEBs) (≥5 PNECs) (**c**) per lung section from each embryo ($n = 5$ embryos with 37 sections counted for $Rest^{+/+}$ and 4 embryos with 30 sections counted for $Rest^{+/fl}$ and $Rest^{fl/fl}$). Unpaired t-test, data represented as mean ± s.d. In (**d**), a scatterplot of all solitary PNECs and clusters of each embryo ($n = 438$ for $Rest^{+/+}$, 688 for $Rest^{fl/fl}$). Wilcoxon rank sum test, two-sided, on the combined data points of each genotype. **e**, Median number of PNECs per cluster for each embryo as in (**d**) ($n = 5$ embryos for $Rest^{+/+}$ and 4 embryos for $Rest^{fl/fl}$). Unpaired t-test, data represented as mean ± s.d. **f** Relative mRNA expression levels of NE markers and $Rest$ in E18.5 $Shh^{cre/+}$;$Rest^{+/+}$, $Rest^{+/fl}$, and $Rest^{fl/fl}$ lungs ($n = 5$ embryos for $Rest^{+/+}$ and 4 embryos for $Rest^{+/fl}$ and $Rest^{fl/fl}$). Unpaired t-test, data represented as mean ± s.d. **g** Representative brightfield images of $Chga$ RNAscope in situ hybridization (ISH, red signal) in E18.5 $Shh^{cre/+}$;$Rest^{+/+}$ and $Rest^{fl/fl}$ lungs (hematoxylin counterstain). Scale bar, 50 µm. **h** Quantification of the average number of RNAscope signal per µm$^2$ of lung section in (**g**) ($n = 3$ embryos for $Rest^{+/+}$ and 4 embryos for $Rest^{fl/fl}$). Unpaired t-test, data represented as mean ± s.d. **i**, Representative images for $Chga$ RNAscope ISH (orange) and E-cadherin immunofluorescence (red) in E18.5 $Shh^{cre/+}$;$Rest^{+/+}$ and $Rest^{fl/fl}$ lungs. DAPI stains the DNA in blue. White arrows, NEBs. Scale bar, 100 µm. See also Supplementary Fig. 8. Source data are provided as a Source Data file.

**Yap1 deletion prevents NE to non-NE transition after injury.** In the adult lung, $Yap1$ and $Wwtr1$ are not expressed in PNECs except for a few rare cells (Supplementary Fig. 10a)[24]. In contrast, PNECS that transdifferentiate to a club cell fate during naphthalene-induced injury express detectable levels of YAP1 (Supplementary Fig. 10b). We therefore wondered if YAP1, like REST, can also modulate cell fate decisions in PNECs in non-cancer contexts. $Yap1$ deletion early in the embryonic lung epithelium in mice leads to a general failure in cell type

specification[53], making it impossible to investigate a specific role for YAP1 in PNEC specification. We thus tested the requirement for $Yap1$ in PNEC transdifferentiation during injury.

We crossed $Yap1^{fl/fl}$ mice to $Ascl1^{CreER/+}$;$Rosa26^{LSL-Tom/LSL-Tom}$ mice and labeled PNECs with Tomato by injecting tamoxifen followed by naphthalene to induce injury. We then allowed the mice a 3-week recovery period before harvesting the lungs for analysis. Immunofluorescence staining of lung sections from these mice identified Tomato-positive cells expressing CC10 in control

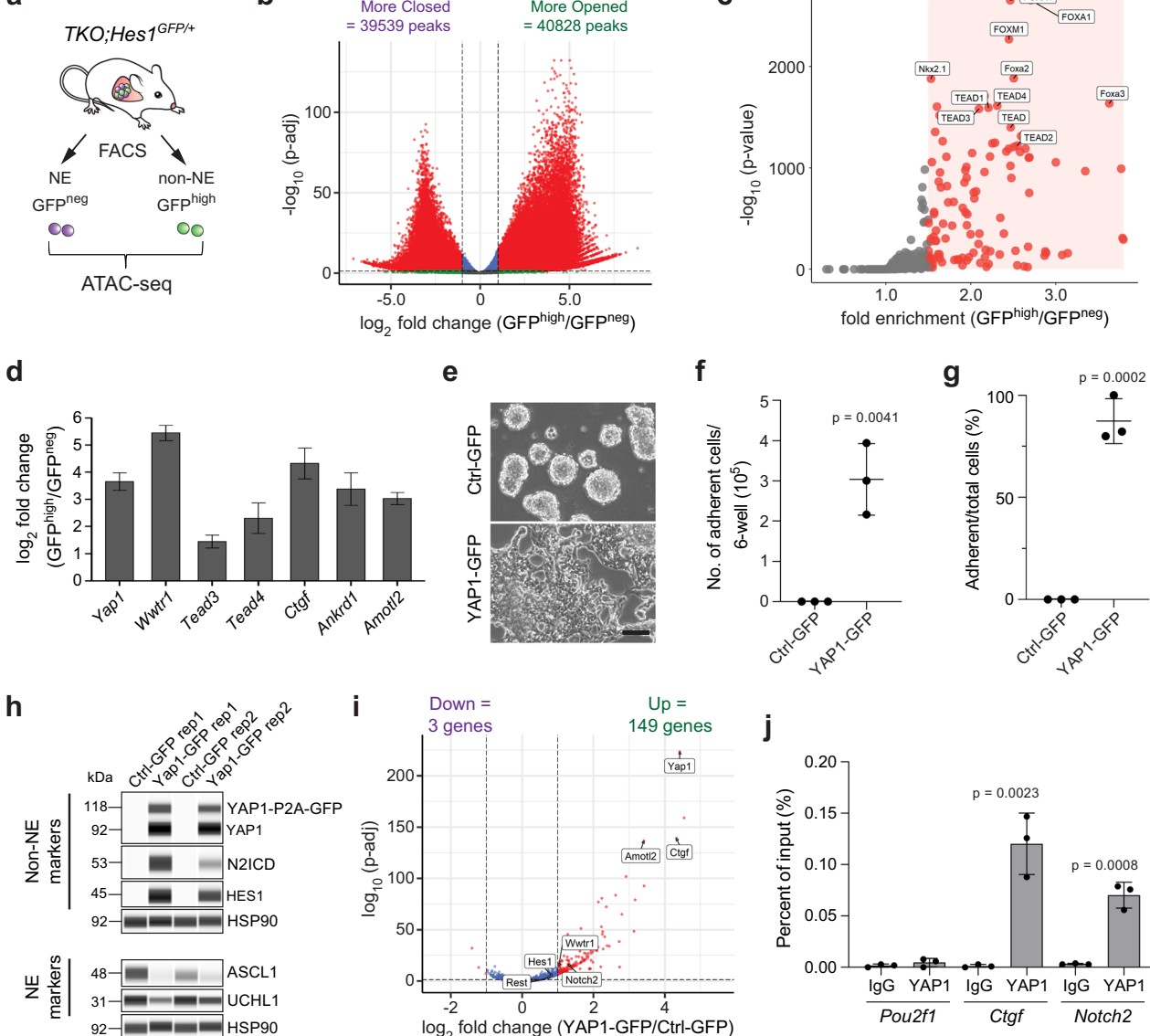

**Fig. 6 YAP1 promotes the transdifferentiation of SCLC cells to a non-neuroendocrine fate. a** Schematic representation of neuroendocrine (NE) and non-NE SCLC cells isolated from the *TKO;Hes1GFP/+* mouse model for ATAC-seq. **b** Volcano plot of differentially accessible peaks from the ATAC-seq data comparing non-NE SCLC cells to NE SCLC cells (significant peaks with more than 2-fold change and *p*-adj value < 0.001 are in red). Wald test with Benjamini–Hochberg correction. **c** Scatterplot showing the enriched motifs identified by HOMER analysis on the more open peaks in non-NE SCLC cells (significant motifs with more than 1.5-fold change and *p*-value < 0.001 are in red). **d** Expression of Hippo pathway members and targets from RNA-seq data in GFPhigh cells compared to GFPneg cells (*n* = 4). Error bars, log2 fold change estimate ± SE. **e–g, e** Representative brightfield images of NE mouse KP1 SCLC cells 14 days after Ctrl-GFP (control) or YAP1-GFP overexpression. Scale bar, 100 μm. Quantification of the number of adherent cells counted per well of a 6-well plate (**f**) and the percentage of cells that became adherent (**g**) as in (**e**) (*n* = 3 independent experiments). Unpaired *t*-test, data represented as mean ± s.d. **h** Immunoassay of non-NE and NE markers in KP1 cells after overexpression of either Ctrl-GFP or YAP1-GFP (4 weeks). Bands of two molecular weights are detected with the antibody against YAP1 due to some incomplete cleavage of P2A. Samples from *n* = 2 independent experiments. **i** Volcano plot of differentially expressed genes from RNA-seq data of 5 days overexpression of YAP1 in KP1 cells. *Notch2*, *Hes1*, *Rest* and Hippo pathway members are highlighted (significant genes with more than 2-fold change and *p*-adj value < 0.05 are in red). Wald test with Benjamini–Hochberg correction. **j** ChIP-qPCR analysis of YAP1 binding at the promoters of *Pou2f1* (negative ctrl), *Ctgf* (positive ctrl) and *Notch2* in independently generated GFPhigh cell lines from *TKO;Hes1GFP/+* mouse SCLC tumors (*n* = 3). Unpaired *t*-test, data represented as mean ± s.d. See also Supplementary Fig. 9 and Supplementary Data 12 and 13. Source data are provided as a Source Data file.

mice but not *Yap1* mutant mice. Furthermore, lineage-labeled cells with CC10 expression also co-expressed YAP1 whereas those that did not transdifferentiate remained YAP1 negative (Fig. 7a, b). In addition, quantification of Tomato-positive clusters in the lungs by whole mount imaging showed reduced numbers of large Tomato-positive clusters in *Yap1* mutant mice compared to controls (Fig. 7c, d). These experiments indicate that *Yap1* is required for

the NE to non-NE transition in response to naphthalene injury in the lung epithelium.

## Discussion

Progress on understanding critical molecular events in Notch-driven cell fate changes in normal tissues has often been limited

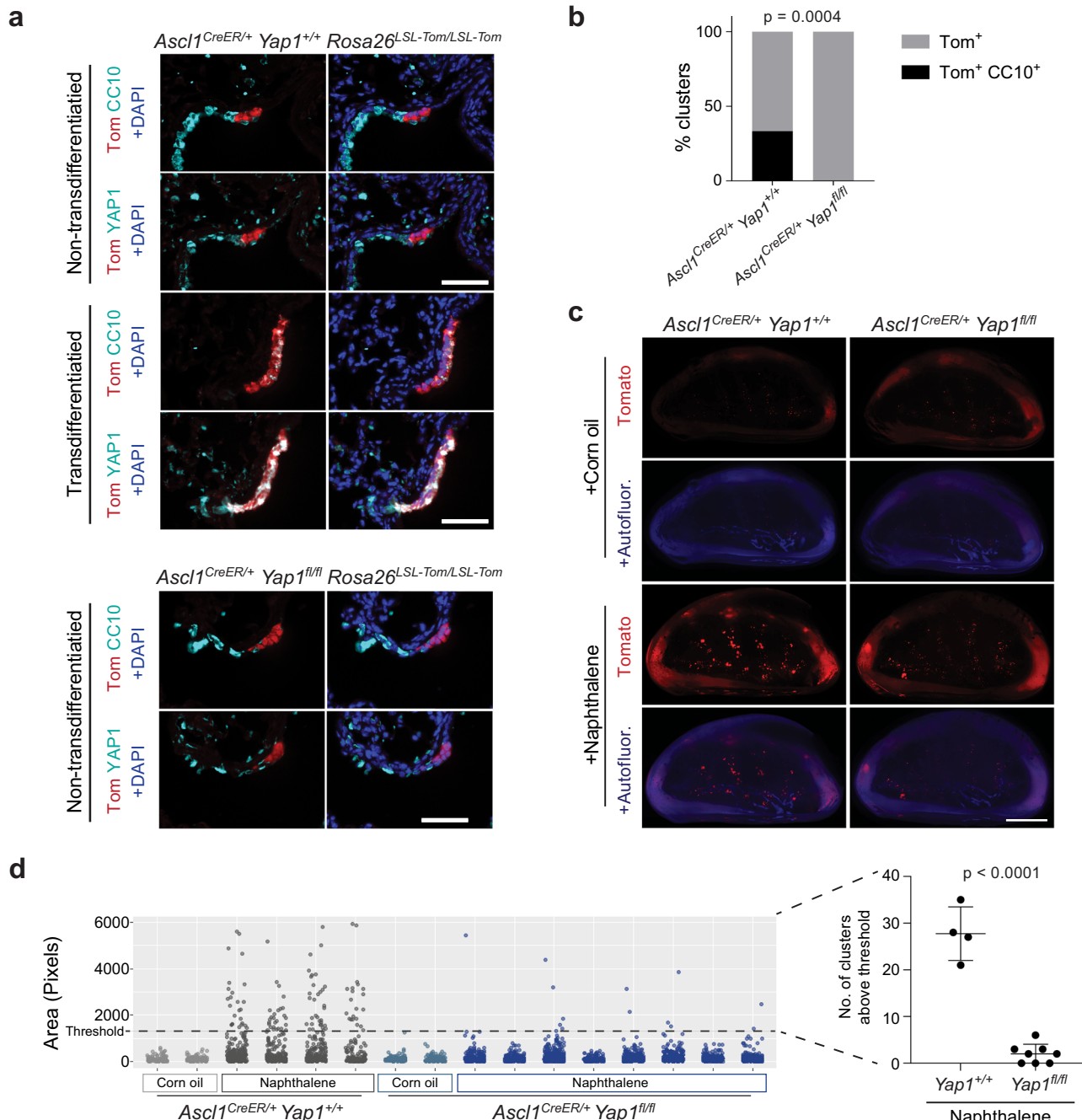

**Fig. 7 YAP1 is required for the transdifferentiation of pulmonary neuroendocrine cells after naphthalene injury. a** Representative immunofluorescence images for Tomato expression (red) and either CC10 or YAP1 (cyan) on consecutive sections from a $Ascl1^{CreER/+}$;$Rosa26^{LSL-Tom/LSL-Tom}$;$Yap1^{+/+}$ or $Yap1^{fl/fl}$ mouse 3 weeks after naphthalene-induced lung injury and tamoxifen injection (as in Fig. 4b). DAPI stains DNA in blue in the images on the right. Scale bar, 50 μm. **b** Quantification (%) of clusters that are Tom+ or Tom+, CC10+ after injury as in (**a**) ($n = 27$ clusters for $Yap1^{+/+}$ and 32 clusters for $Yap1^{fl/fl}$ counted from 3 mice per genotype). Fisher's exact test, two-tailed. **c** Whole mount fluorescence images of left lung lobes 3 weeks after injury. Images representative of $n = 2$ independent experiments. Scale bar, 5 mm. **d** Scatterplot of normal NEBs and PNEC outgrowths after injury as in (**c**) with each column in the scatterplot representing a single mouse and each data point a tomato-positive cluster in that mouse ($n = 2$ mice for corn oil controls, 4 mice for naphthalene-treated $Yap1^{+/+}$ and 8 mice for naphthalene-treated $Yap1^{fl/fl}$). Unpaired $t$-test was conducted on the number of clusters from each mouse in naphthalene-treated groups that are above the threshold size of the non-injured controls ($n = 4$ mice for $Yap1^{+/+}$, $n = 8$ mice for $Yap1^{fl/fl}$), data represented as mean ± s.d. See also Supplementary Fig. 10. Source data are provided as a Source Data file.

by the rarity of the cells involved in these transitions in vivo and the challenges associated with culturing these cells. We show here that SCLC hijacks normal developmental pathways and that findings relevant to Notch signaling in this cancer model where cells are not limiting can be extrapolated to normal tissue development and injury. Using SCLC, we

identified REST and YAP1 as two additional important contributors to Notch-driven regulation of pulmonary neuroendocrine cell fate decisions. Both REST and YAP are antagonistic to the neuroendocrine fate and when lost, promotes NE specification or impedes transition towards the non-NE fate depending on the context (Fig. 8a, b).

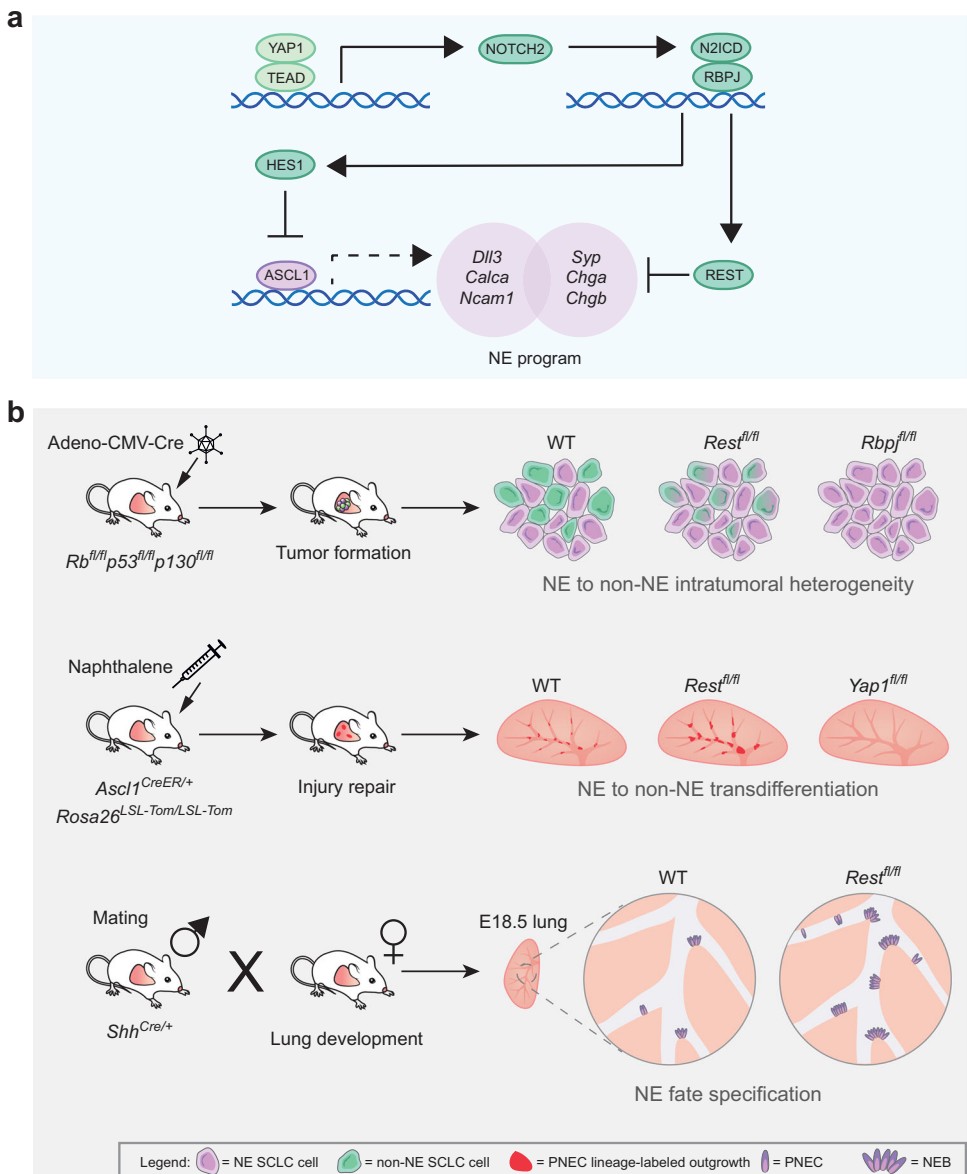

**Fig. 8 A conserved YAP/Notch/REST network controls the neuroendocrine cell fate in the lungs. a** Schematics of the molecular pathway governing the activation/repression of the neuroendocrine programs. Active YAP1 upregulates the expression of NOTCH2, which in turn increases the expression of REST and HES1. HES1 represses the transcription of *Ascl1*, indirectly repressing the genes under ASCL1 transcriptional control, while REST directly represses the transcription of a distinct part of the neuroendocrine program. **b** Schematics of the in vivo experiments testing the effect of the loss of *Rest/Rbpj/Yap1* on intratumoral heterogeneity, transdifferentiation during injury repair, and neuroendocrine fate specification during lung development. In the tumor model, loss of *Rest* prevents full transition to the non-neuroendocrine fate in contrast to *Rbpj*, which completely blocks the transition. In the injury model, *Rest* deletion delays terminal transdifferentiation resulting in a longer proliferation phase and larger outgrowths. *Yap1* deletion blocks the ability of PNECs to respond to the injury, leading to the reduction in outgrowths/transdifferentiation. In the development model, the absence of REST promotes the neuroendocrine fate, causing an increase in the number of pulmonary neuroendocrine cells (PNECs) and neuroepithelial bodies (NEBs) in the lung.

Studies on PNEC specification in early development and transdifferentiation after injury have centered mainly around NOTCH receptors/ligands and ASCL1. The loss of NE signature when Notch signaling is activated has been attributed to the decrease in expression and activity of ASCL1, a well-known transcriptional activator of NE genes[12,54–56]. However, our previous work showed that knock-down of ASCL1 in NE SCLC cells in culture is not sufficient to switch these cells to a non-NE fate and placed the transcriptional repressor of neuronal genes, REST, downstream of Notch[29]. Our new data reveal that REST and ASCL1 govern largely independent transcriptional programs determining NE fate. In the absence of REST, a large number of

NE genes cannot be repressed or show significantly delayed repression as cells switch fate towards a non-NE state. These experiments conclusively demonstrate that REST is an important mediator of Notch signaling to promote the non-NE fate. Recent work in the pancreas[57] indicates that this role for REST in controlling neuroendocrine differentiation is conserved in other lineages.

Based on our findings, we place YAP1 upstream and REST downstream of Notch signaling. However, we cannot exclude the possibility of different sequences of events in different scenarios or the existence of feedback loops. For example, we observed upregulation of *Notch2* after REST overexpression in SCLC cells

in our RNA-seq data. *Rest* expression may also be induced by YAP1 in mechanisms independent of Notch signaling[38]. There is further evidence of crosstalk and bidirectional regulation between the Hippo/YAP and Notch pathways in other settings[58,59]. In an example directly relevant to our study, Yao and colleagues observed nuclear YAP1 in lineage-labeled PNECs expressing constitutively active NICD 3 days post-naphthalene injury[23]. Assuming that YAP1 is downstream of Notch signaling based on these data, the authors deleted *Yap1* in NICD-expressing cells and observed no changes in the phenotypes analyzed, which led them to conclude that Hippo pathway was not required[23]. Yet it remains possible that YAP1 is upstream of Notch signaling initially, but that the two pathways work together in a positive feedback loop to drive the transdifferentiation process[38].

By single-cell RNA-seq analysis, most of the PNECs do not express *Yap1* except for a few rare cells[24]. Interestingly, only 20–30% of NEBs form "outgrowths" after naphthalene injury, and it has been postulated that these outgrowths arise from a rare type of "stem-cell like" PNECs that express *Notch2*[24]. Other than naphthalene-induced injury, another commonly studied lung injury is hypoxia. Studies in breast cancer have shown that hypoxia can inhibit Hippo signaling leading to YAP1 accumulation in the nucleus[60]. In the lungs, hypoxia induces proliferation in ~20% of the NEBs[61] similar to the 20–30% noted in naphthalene-induced lung injury[24]. It is tempting to conjecture that the subset of NE cells induced to proliferate by hypoxia and naphthalene is the group of "stem-cell like" NE cells that may express *Yap1*. A simple model would then be that *Yap1* and *Notch2* are co-expressed in the same "stem-cell like" PNECs. The current single-cell RNA-seq data do not support this idea, although this may be due to detection limits of the technology. Dual lineage-tracing systems to label *Ascl1* and *Yap1* expressing PNECs will be needed to address this question.

SCLC is classified into four major subtypes based on expression of four key transcription regulators ASCL1, NEUROD1, POU2F3, and YAP1[34], although there has been some debate whether YAP1 expression marks a true subtype or is a reflection of intratumoral heterogeneity[62–64]. Tumors in our *TKO* mouse model are of the ASCL1 subtype and based on our findings here, YAP1-expressing cells are non-NE and part of the intratumoral heterogeneity. Hence, SCLC subtype classification based on YAP1 alone may not be sufficient. We previously published that Notch-active non-NE cells, while growing slower than their counterpart, secrete factors to promote the growth of the NE cells; these cells are also more resistant to chemotherapy[29]. High expression of NOTCH1 correlates with decreased survival in SCLC patients[65]. Recent evidence further indicates that YAP1-driven reprogramming of SCLC cells towards a non-NE phenotype can decrease the sensitivity of SCLC cells to chemotherapy[38]. Thus, it might be beneficial to target these non-NE cells. Unfortunately, the NOTCH2/3 inhibitor Tarextumab in combination with chemotherapy failed a phase 2 SCLC clinical trial in 2017 with reasons unclear as no biopsies were taken[66]. In addition, SCLC tumors comprise very few non-NE cancer cells when they metastasize in mouse models[29], suggesting that the non-NE cells are not absolutely required during SCLC progression. Furthermore, YAP/TAZ may overall be tumor suppressors in SCLC[67], which makes it unclear whether YAP1⁺ non-NE cells should be targeted or not. Nevertheless, one main reason for fast therapy resistance/relapse is attributed to the plasticity of SCLC cells which might be alleviated by limiting transdifferentiation[37,64,68]. A possible therapeutic strategy would be to use activators or inhibitors of TEAD and REST[69–71] to drive the cells to a non-NE fate or lock the cells in a NE fate in combination with other therapies, for example ASCL1 inhibitors[72] or Notch pathway regulators[17,56] that exploit differential vulnerabilities of the various cell states in SCLC.

Our identification of common transcriptional regulators and pathways in PNEC fate decisions and SCLC transdifferentiation underscores the relevance of PNEC biology to the evolution of SCLC intratumoral heterogeneity and vice versa. Discoveries in both fields will aid in development of better treatment options for not just SCLC but also other diseases linked to the activity of PNEC, including asthma and other immune-related activities in the lungs[73,74].

## Methods

**Ethics statement**. Mice were maintained according to practices prescribed by the NIH at Stanford's Research Animal Facility (protocol #13565). Additional accreditation of Stanford animal research facilities was provided by the Association for Assessment and Accreditation of Laboratory Animal Care (AAALAC).

**Animal studies**. Mouse lines used were the triple-knockout (*TKO*) SCLC mouse model bearing deletions of floxed (fl) alleles of *p53*, *Rb*, and *p130* as previously described[75], GFP reporter expressed from the endogenous *Hes1* promoter *Hes1^GFP* allele[76], Tomato reporter *Rosa26^lox-stop-lox-tdTomato*[77], GFP reporter under the control of *Chga* promoter *Chga-GFP* (inserted as multiple copies of BAC, also crossed to *TKO* but no Cre was introduced)[44], floxed *Yap1* allele *Yap1^fl*[78], floxed *Rest* allele *Rest^fl* (also named *Rest^GTi*)[39], floxed *Rbpj* allele *Rbpj^fl*[79], Cre recombinase knocked into *Ascl1* locus *Ascl1^CreER* allele (also named *Ascl1^CreERT2*)[80] and Cre recombinase knocked into *Shh* locus *Shh^Cre* allele (also named *Shh^GFPcre*, featuring Cre fused to GFP)[81]. *Dner* KO mice were generated by the Stanford Transgenic, Knockout, and Tumor Model Center (TKTC) via injection of C57BL/6 mouse zygotes with sgRNAs and Cas9 targeting sites in intron 1 and 4, resulting in a deletion of exon 2–4 containing the delta/notch-like EGF repeat region.

For in vivo SCLC studies, tumors were induced in 8–12 weeks old mice of mixed gender by intratracheal instillation with $4 \times 10^7$ PFU of Adeno-CMV-Cre (Baylor College of Medicine, Houston, TX)[29]. Mice were euthanized before or at the first sign of respiratory distress in accordance with Stanford APLAC guidelines. Mice were housed at 22 °C ambient temperature with 40% humidity and a light/dark cycle of 12 h (7 a.m.–7 p.m.).

**Flow cytometry sorting of cells**. Tumors isolated from the lungs of *TKO;Hes1^GFP/+* mice approximately 6–7 months after tumor induction were pooled, finely chopped with a razor blade, and digested in 6 mL of PBS with 120 μL of 100 mg/mL collagenase/dispase (Roche) for 45 min in a 37 °C shaker. The mixture was then cooled on ice before adding 15 μL of 1 mg/mL DNase (Sigma) for 5 min. Digested tissue was passed through a 40 μm filter, spun down, and resuspended in 1 mL of red blood cell lysis buffer (150 mM $NH_4Cl$, 10 mM $KHCO_3$, 0.1 mM EDTA) for 1.5 min. Cells were spun down, washed once in DMEM, and resuspended in FACS buffer (10% BGS in PBS, 1 million cells per 100 μL). The resultant single-cell suspension was stained with the following FACS antibodies: CD45-PE-Cy7 (eBioscience 25-0451-82, clone 30-F11, 1:100), CD31-PE-Cy7 (eBioscience 25-0311-82, clone 390, 1:100), TER-119-PE-Cy7 (eBioscience 25-5921-82, clone TER-119, 1:100), CD24-APC (eBioscience 17-0242-82, clone M1/69, 1:200) or CD45-Pacific Blue (BioLegend 103126, clone 30-F11, 1:100), CD31-Pacific Blue (BioLegend 102422, clone 390, 1:100), TER-119-Pacific Blue (BioLegend 116232, clone TER-119, 1:100), CD24-PE-Cy7 (Biolegend 101821, clone M1/69, 1:200), ICAM1/CD54-PerCP-Cy5.5 (Biolegend 116123, clone YN1/1.7.4, 1:100), NCAM1/CD56-APC (R&D Systems FAB7820A, clone 809220, 1:50) and DAPI to label dead cells.

Lungs from *Chga-GFP* mice were digested and stained as above with an additional lineage negative selection marker F4/80-PE-Cy7 (BioLegend 123114, clone BM8, 1:100) and EpCAM-APC (eBioscience 17-5791-82, clone G8.8, 1:100) instead of CD24 as positive selection marker.

Tumors isolated from the lungs of *TKO;Rest^fl/fl* or *TKO;Yap1^fl/fl* mice were digested in the same way but stained with the following FACS antibodies: CD45-Pacific Blue (BioLegend 103126, clone 30-F11, 1:100), CD31-Pacific Blue (BioLegend 102422, clone 390, 1:100), TER-119-Pacific Blue (BioLegend 116232, clone TER-119, 1:100), CD24-APC (eBioscience 17-0242-82, clone M1/69, 1:200), NCAM1-Alexa Fluor 488 (R&D Systems FAB7820G, clone 809220, 1:50), and ICAM1-PE-Cy7 (BioLegend 116122, clone YN1/1.7.4, 1:100).

FACS was performed with a 100 μm nozzle on a BD FACSAria II using the FACSDiva software. Data were analyzed and plotted using FlowJo v10.

**Genotyping**. DNA was extracted from cells using the Allprep DNA/RNA microkit (Qiagen 80284). 1 ng of DNA was used for each PCR reaction to detect unrecombined (floxed) and recombined (delta, Δ) *Rb*, *p53*, and *p130* alleles. Primer sequences are available in Supplementary Data 14.

**RNA-seq library preparation, sequencing, and analysis**. $2 \times 10^5$ cells were sorted for each sample from tumors of *TKO;Hes1^GFP/+* mice, REST or YAP1 over-expressing cell lines in respective experiments. RNA was isolated using Allprep DNA/RNA microkit (Qiagen 80284) according to manufacturer's protocol and concentration determined by Agilent Bioanalyzer 2100. Libraries were prepared

using TRIO RNA-seq kit (Nugen 0507-32) according to manufacturer's protocol with 5 ng of RNA for in-vivo tumor cells or 10 ng of RNA for in-vitro cell lines per sample as starting material. Concentration of libraries were determined using KAPA library quantification kit (KAPA Biosystems KK4835) and samples were pooled evenly for sequencing using Illumina Nextseq 500 or Hiseq 4000 to obtain approximately 30 million paired-end reads per sample.

$5 \times 10^6$ cells were pelleted and snap frozen for each cell line generated from tumors of *TKO* or *TKO;Rest*<sup>fl/fl</sup> mice. RNA isolation, library preparation and sequencing were performed by Novogene using Illumina platforms to obtain approximately 30 million paired-end reads per sample.

For the *Ascl1* knock-down experiments, RNA was isolated from sh*Ascl1*-transduced cell lines using the Allprep DNA/RNA microkit (Qiagen 80284) and sent to Novogene for library preparation and sequencing as described above.

Transcript count files for RNA-seq data were generated using Salmon[82] (combine-lab.github.io/salmon/). Differential expression analysis was conducted using DESeq2[83] with default settings. Plots are generated using ggplot2[84] (https://ggplot2.tidyverse.org/) and EnhancedVolcano[85] (github.com/kevinblighe/EnhancedVolcano).

**Gene set enrichment analysis.** Normalized count files generated by DESeq2 were used as input for Gene set enrichment analysis[86,87] (GSEA). GSEA was performed using MSigDB Hallmark gene sets with the following settings (Number of permutations = 1000, Permutation type = gene_set).

**Gene ontology enrichment analysis.** Gene ontology (GO) enrichment analysis was performed using Metascape[88] with default settings in the "Express Analysis" or "Custom Analysis: Cellular components/Biological Processes" functions.

**Single-cell fluidgm analysis.** Single cells were sorted into individual wells of a 96-well PCR plate containing 5 μL of 2× reaction mix (CellsDirect<sup>TM</sup> One-Step RT-qPCR kit, Invitrogen 11753) and 2 units of SUPERase-In RNase inhibitor (Ambion, AM2696). Plates are stored at −80 °C till processing. Primers were designed using Fluidigm D3 assay design system and pooled to a final concentration of 500 nM to be used at 10× dilution in the PCR reaction. One-step reverse transcription and pre-amplification were performed on the sorted cells with the pooled primers and SuperScript III RT/Platinum *Taq* Mix from the kit using the following cycle conditions: 15 min 50 °C, 2 min 95 °C, 20 cycles of 15 s 95 °C and 4 min 60 °C, 15 min 4 °C. The cDNA products were treated with Exonuclease I (NEB M0293S) for 30 min at 37 °C, inactivation at 80 °C for 15 min to remove excess primers and diluted 5-fold with low EDTA TE (Affymetrix 75793) for the final reaction. 2.25 μL of diluted cDNA, 2.5 μL 2× Sso Fast EvaGreen Supermix with low ROX (Bio-Rad #172-5211) and 0.25 μL 20× DNA binding dye sample loading reagent (Fluidigm #100-3738) were mixed and then loaded into a 48.48 Dynamic Array<sup>TM</sup> integrated fluidic circuit (IFC) chip sample inlets. 0.25 μL of each 100 μM primer pair, 2.5 μL 2× Assay Loading reagent (Fluidigm #85000736) and 2.25 μL low EDTA TE were mixed and loaded into the IFC assay inlets. The chip was ran on a Biomark<sup>TM</sup> machine according to the manufacturer's protocol for EvaGreen probes. Primer sequences are available in Supplementary Data 14.

**Single-cell RNA-seq analysis.** Single-cell RNA-seq TPM data of PNECs and other lung epithelial cell types in a normal lung were downloaded from GEO:GSE136580[24].

**Immunostaining.** For immunofluorescence on paraffin sections, tissues were first fixed overnight in 10% neutral buffered formalin (NBF) and processed for paraffin embedding. Paraffin sections were deparaffinized with Histo-clear (National Diagnostics HS-200) and rehydrated in decreasing strengths of ethanol and finally water. Antigen retrieval was carried out in citrate-based unmasking solution (Vector Laboratories H-3300) by heating in the microwave till boiling followed by additional 12 min at 30% power. Sections were washed in PBST (PBS with 0.1% Tween-20) and blocked in 5% horse serum for 1 h. Primary antibodies diluted in PBST with 5% horse serum were then added to the sections for incubation overnight at 4 °C.

To co-stain YAP1 with CC10 in tumors, staining of YAP1 was done first by alkaline phosphatase-based method using Impress®-AP anti-rabbit IgG kit (Vector Laboratories MP-5401) following the manufacturer's protocol and color development with ImmPACT® Vector® Red alkaline phosphatase substrate (Vector Laboratories SK-5105), which fluoresces in the red channel. The sections were then reblocked for an hour in 5% horse serum before incubating with anti-CC10 primary antibody overnight at 4 °C and fluorophore-conjugated secondary antibody for an hour at room temperature the next day. Nuclei were counterstained with DAPI (Sigma), and slides mounted with Fluoromount-G® (Southern Biotech 0100-01). The following antibodies were used: Yap(D8H1X)XP® (Cell Signaling Technology 14074S, 1:200), CC10/Scgb1a1 (Santa Cruz sc-9772, 1:50) and Alexa Fluor®488 donkey anti-goat IgG (Invitrogen A-11055, 1:200).

To co-stain SYP or YAP1 with CC10 in the injury models, CGRP with tdTomato in the injury models, RPB-J or HES1 with CC10 in tumors, paraffin sections were deparaffinized and heat-antigen retrieved as above. Endogenous peroxidase was blocked by incubating slides in 3% hydrogen peroxide. Slides were

then washed in PBST, blocked in 5% horse serum and incubated with primary antibodies overnight at 4 °C. SYP, YAP1, CGRP, RBP-J or HES1 was developed first by using ImmPRESS® HRP Horse Anti-Mouse IgG Polymer Detection Kit (Vector Laboratories MP-7402) for SYP or ImmPRESS® HRP Horse Anti-Rabbit IgG Polymer Detection Kit (Vector Laboratories MP-7401) for CGRP, YAP1, RBP-J and HES1 following the manufacturer's protocol and color development with TSA Plus Fluorescein (Akoya Biosciences). Sections were then washed in PBST and incubated with fluorophore-conjugated secondary antibody against CC10 primary antibody for an hour at room temperature. Tomato was stained in the same manner as CC10. Nuclei were counterstained with DAPI (Sigma), and slides mounted with Fluoromount-G® (Southern Biotech 0100-01). The following antibodies were used: Syp (Neuromics MO20000, 1:200), Yap(D8H1X)XP® (Cell Signaling Technology 14074S, 1:200), RBP-J/RBPSUH (D10A4)XP® (Cell Signaling Technology 5313, 1:200), HES1 (Cell Signaling Technology 11988, 1:200), CGRP (Sigma-Aldrich C8198, 1:200), CC10/Scgb1a1 (Millipore 07-623, 1:200), CC10/Scgb1a1 (Santa Cruz sc-9772, 1:50), RFP/Tomato (Biosource MBS448092, 1:500), Alexa Fluor®594 donkey anti-rabbit IgG (Invitrogen A-21207, 1:200), Alexa Fluor®488 donkey anti-rabbit IgG (Invitrogen A-21206, 1:200) and Alexa Fluor®594 donkey anti-goat IgG (Invitrogen A-11058, 1:200).

For immunofluorescence on cryosections, in embryo studies, pregnant mice were injected with 100 mg/kg of BrdU (Sigma B5002) 2 hrs before sacrificing. E18.5 embryos were dissected out with the left lung lobe taken for immediate freezing in O.C.T. (Sakura Tissue-Tek® O.C.T.<sup>™</sup> Compound 4583) with liquid nitrogen into cryoblocks that were kept at −80 °C till sectioning. Tumor-bearing lungs from adult *TKO* mice were processed in the same manner as above with an additional step of lung inflation with 50% O.C.T in PBS before embedding. The cryoblocks were sectioned at 7 μm thickness. For quantification of PNECs in embryonic lungs, 1 section was collected approximately every 4 sections. The cryosections were kept in an airtight tube at −80 °C till staining. Cryosections were equilibrated to room temperature in the tube before being taken out to fix in 4% paraformaldehyde for 10 min. Slides were washed once in PBS and once in distilled water for 5 min each. Antigen retrieval was carried out in citrate-based unmasking solution (Vector Laboratories H-3300) by heating in the microwave till boiling and then immediate cooling on the benchtop for 30 min. Sections were blocked in blocking solution (5% horse serum, 0.25%Triton X-100) for an hour at room temperature before incubation with primary antibodies overnight at 4 °C. The following day, sections were washed three times for 5 min each in PBS followed by secondary antibody incubation for an hour at room temperature and nuclei counterstain with DAPI. Finally, slides were mounted with Fluoromount-G® (Southern Biotech 0100-01) for viewing. The following antibodies were used: CGRP (Sigma-Aldrich C8198, 1:200), BrdU (BD Biosciences 347580, 1:50), Dner (R&D Systems AF2254, 1:50), AGER/RAGE (R&D Systems MAB1179, 1:200), CC10/SCGB1A1 (Millipore 07-623, 1:200), Alexa Fluor®594 donkey anti-rabbit IgG (Invitrogen A-21207, 1:200), Alexa Fluor®488 donkey anti-rabbit IgG (Invitrogen A-21206, 1:200), Alexa Fluor®594 donkey anti-rat IgG (Invitrogen A-21209, 1:200) and Alexa Fluor®488 donkey anti-mouse IgG (Invitrogen A-21202, 1:200).

For immunohistochemistry (DAB) on paraffin sections, slides were deparaffinized and heat-antigen retrieved in the same manner as immunofluorescence. Endogenous peroxidase was blocked by incubating slides in 3% hydrogen peroxide. Slides were then washed in PBST, blocked in 5% horse serum, and incubated with primary antibodies overnight at 4 °C. CC10 was developed by using ImmPRESS® HRP Horse Anti-Rabbit IgG Polymer Detection Kit (Vector Laboratories MP-7401) following the manufacturer's protocol and color development with DAB substrate kit (Vector Laboratories SK-4100). HES1 or YAP1 was developed using ImmPRESS® Excel Amplified Polymer Staining Anti-Rabbit IgG Peroxidase kit (Vector Laboratories MP-7601) following the manufacturer's protocol. Sections were counterstained with hematoxylin (Sigma-Aldrich HHS32-1L), dehydrated in increasing strengths of ethanol ending in xylene, and mounted with Refrax mounting medium (Anatech Ltd 711). The following antibodies were used: Hes1 (Cell Signaling Technology 11988, 1:200), Yap(D8H1X)XP® (Cell Signaling Technology 14074S, 1:200), RBP-J/RBPSUH (D10A4)XP® (Cell Signaling Technology 5313, 1:200), CC10/Scgb1a1 (Millipore 07–623, 1:5000).

Images were taken using a Keyence BZ-X700 all-in-one fluorescence microscope with BZ-X viewer program version 1.3.1.1 and BZ-X Analyzer 1.4.0.1.

**DNA/RNA extraction, cDNA synthesis and RT-qPCR analyses.** The right cranial lobe and the right middle lobe of E18.5 embryos were collected, washed in cold PBS, and stored at −80 °C till processing. Cells from in-vitro experiments were counted (200k or 1 million cells), washed once in cold PBS before freezing down as cell pellets. DNA and RNA were extracted using Qiagen Allprep DNA/RNA microkit according to the manufacturer's protocol. 1 μg of RNA was taken as starting material for cDNA synthesis using ProtoScript® First Strand cDNA synthesis (NEB E6300L) with the provided randomized primer mix. The cDNA product was diluted 1:20 with water for RT-qPCR using Perfecta® SYBR® Green Fast Mix (Quanta Biosciences 95073) on a Bio-Rad CFX384 Touch Real-Time PCR Detection System. Data were normalized to *Gapdh* as the housekeeping gene. Primer sequences are available in Supplementary Data 14.

**Naphthalene-induced lung injury.** 8–10 weeks old mice of mixed gender were injected with 1.5 mg of tamoxifen (20 mg/mL in corn oil, Sigma-Aldrich T5648) i.p.

daily for three consecutive days. Cages were changed 72 h later to allow tamoxifen to wash out completely. 7 days after the last tamoxifen injection, mice were injected with a single dose of 200 mg/kg of naphthalene (50 mg/mL in corn oil, Sigma-Aldrich 184500) to induce injury. 3 weeks after naphthalene injection, lungs were harvested and fixed in cold 4% paraformaldehyde (diluted with PBS from 32% paraformaldehyde, Electron Microscopy Sciences 15714) with rocking at 4 °C for 4–5 h, protected from light to preserve endogenous fluorescence. Fixed lungs were washed once in PBS for 2 h followed by a second PBS wash overnight at 4 °C. After which, they were transferred into Cubic-L (Tokyo Chemical Industry T3740) for 4 days (with one change of solution after 2 days) for clearing at room temperature with slow rocking. The lungs were then washed twice in PBS, 2 h each at room temperature, before storing in Cubic-R+ (Tokyo Chemical Industry T3741) at 4 °C for the rest of the time (1–2 days later) till imaging. For whole mount imaging, cleared lungs were placed in inverted coverglass chambers (Nunc Lab Tek 155361) and imaged with ×2 objectives using Keyence BZ-X700 all-in-one fluorescence microscope with BZ-X Viewer program version 1.3.1.1 and BZ-X Analyzer 1.4.0.1. Images were analyzed and quantified using SCF-MPI-CBG Interactive H-watershed plugin on FIJI (Fiji Is Just ImageJ)[89]. Statistical analyses for these injury experiments were performed on clusters containing cells over a threshold (based on corn oil-treated mice) because only a small fraction of neuroendocrine cells is expected to respond to injury[24]. Statistical analysis using median values showed no significance, as shown in the panels.

### Tumor number/area, HES1/CC10/YAP1 DAB and fluorescence staining quantification.

For tumor burden quantification, H&E sections were imaged with ×2 objectives using Keyence BZ-X700 all-in-one fluorescence microscope with BZ-X Viewer program version 1.3.1.1 and BZ-X Analyzer 1.4.0.1. Tumors were manually demarcated as region of interests (ROIs) and tumor number/area measured with Fiji Is Just ImageJ (FIJI). For Hes1 and CC10 DAB staining quantification, tumor regions were imaged with ×20 objectives using the Keyence microscope. Hes1 nuclear staining and CC10 cytoplasmic staining were quantified using ImmunoRatio and ImmunoMembrane plugins on FIJI, respectively[90,91]. HES1 and CC10 fluorescence staining were quantified using the cell detection feature on Qupath[92].

### ChIP-qPCR.

Cells were washed once in PBS and first fixed with 2 mM disuccinimidyl glutarate (Thermo Scientific 20593) in PBS for 30 min to crosslink YAP1 to TEAD followed by a second fixation step with 1% methanol-free formaldehyde in PBS for 10 min at room temperature. Glycine is added to a final concentration of 0.125 M for 5 min to stop the fixation. Cold PBS was added, and cells scraped off into tubes and separated into aliquots of 30 mg pellets for immediate use or snap frozen and stored in −80 °C. Each pellet was resuspended on ice in 1 mL of Farnham lysis buffer (with protease inhibitor) (Boston Bioproducts CHP-110, Roche 05-892-970-001) and passed through a 21-gauge needle 20 times to result in a crude nuclei preparation which was again spun down and resuspended in 500 μL of RIPA buffer (with protease inhibitor) (Boston Bioproducts BP-115). The lysate was then sonicated in 30 s ON/30 s OFF cycles using Diagenode Bioruptor®Plus to obtain chromatin fragments of approximately 150–300 bp. The supernatant was collected and incubated with Yap(D8H1X)XP® (Cell Signaling Technology 14074S) or control IgG (Cell Signaling Technology 2729S) antibodies overnight at 4 °C. The next day, 100 μL of pre-blocked protein A/G magnetic beads (in PBS/BSA) (Pierce 26162) were added and the mixture further incubated for 4 h at 4 °C. The immunoprecipitated chromatin then underwent a series of washes (3 min for each wash at 4 °C): 2 times low salt wash buffer (Boston Bioproducts CHP-160), 3 times high salt wash buffer (Boston Bioproducts CHP-165), 4 times 0.25 M LiCl wash buffer (Boston Bioproducts CHP-155), 1 time TE buffer (Invitrogen 12090-015). Finally, the chromatin was eluted by resuspending the bead pellet in 200 μL of IP elution buffer (Boston Bioproducts CHP-170) at 65 °C for an hour, vortexing every 15 min. Supernatant was collected and incubated at 65 °C overnight to complete reversal of crosslinks. 1 μL of RNase A (Invitrogen 12091-021) was added per de-crosslinked sample for 30 min at 37 °C followed by 4 μL of 0.5 M EDTA, 8 μL of 1 M Tris–HCl (pH 8.0) and 2 μL of Proteinase K (New England Biolabs P8107S) for 2 h at 55 °C. The immunoprecipitated DNA was then purified using QIAquick PCR Purification kit (Qiagen 28106) and eluted in 60 μL water for downstream qPCR, 1 μL per qPCR reaction. Primer sequences are available in Supplementary Data 14.

### ChIP-seq and analysis.

For REST ChIP-seq (two biological replicates), ChIP was performed as above with REST antibody (Millipore 17-641) skipping the 2 mM disuccinimidyl glutarate fixation step. ChIP-seq libraries were constructed using the Ovation Ultralow System V2 kit (Nugen 0344NB-32) according to manufacturer's protocol and sequenced on Illumina Hiseq4000 to obtain single-end reads of 75 bp. For ASCL1 ChIP-seq, fastq data were downloaded from GEO:GSE69394 (one replicate)[16]. ChIP data were aligned using BWA[93] and Samtools[94] to the mouse genome mm10 using the default settings. Aligned sequences were then processed using HOMER[95] (http://homer.ucsd.edu/homer/) using the default settings (TAG directories, finding peaks, and motif analysis). Fold change is relative to input. Heatmap of binding sites were generated with default settings using deepTools[96].

### Cell cultures and cell lines.

Mouse SCLC cell lines were grown in RPMI-1640 medium (Corning MT15040CV) and 293T in DMEM medium (Hyclone SH30243.01) supplemented with 10% bovine growth serum (Hyclone SH3054103HI) and penicillin–streptomycin–glutamine (Gibco 10378016). Mouse KP1 SCLC cell line was generated in the lab and been described[97]. All other cell lines were isolated by FACS from the tumors of individual mice. 293T cells were purchased from ATCC. All cell lines tested negative for mycoplasma.

### REST and YAP1 overexpression in vitro studies.

Control-P2A-GFP, mRest-P2A-GFP and hYap1-P2A-GFP were cloned into a lentiviral pCDH-CMV-MCS vector backbone (Addgene plasmid #72265) and packaged in 293T cells by co-transfection with delta8.2 and VSV-G using PEI. Virus supernatant was collected 48 h after transfection and added to KP1 cells. KP1 cells were then sorted 48 h or 5 days after virus transduction for RNA-seq. For Yap1 transdifferentiation assay, cells were sorted on day 5 and $2 \times 10^5$ cells seeded per 6-well with culture medium refreshed every 4 days. At the end of day 14, the number of floating and adherent cells were trypsinized and quantified.

### Ascl1 knockdown in vitro studies.

Short hairpin RNAs against Ascl1 (#1: 5′-CTCCAACGACTTGAACTCTAT-3′; #2: 5′-CCACGGTCTTTGCTTCTGTTT-3′) from the MISSION shRNA library (Sigma-Aldrich) were packaged in 293T cells by co-transfection with delta8.2 and VSV-G using PEI-based transfection. Virus supernatant was collected 48 h after transfection to infect KP1 cells. Puromycin (1 μg/mL) was added 24 h later for selection. The transduced cells were then harvested on Day 3 after transduction for RNA-seq analysis.

### ATAC-seq library preparation and analysis.

$5 \times 10^4$ cells were sorted for each Hes1-GFP high or negative sample from pooled tumors of each individual $TKO;Hes1^{GFP/+}$ mouse. ATAC-seq libraries were constructed as described in ref.[49]. Briefly, cells were washed once in cold PBS buffer and resuspended in 50 μL of cold lysis buffer (10 mM Tris–HCl, pH 7.4, 10 mM NaCl, 3 mM MgCl₂ and 0.1% IGEPAL CA-630) and centrifuged immediately for 10 min at $500 \times g$, 4 °C to generate a crude nuclei preparation. Nuclei pellets were resuspended in transposition mix (Illumina FC-121-1030: Nextera kit components—25 μL 2× TD buffer, 2.5 μL TDE1 transposase and 22.5 μL nuclease-free water) and incubated at 37 °C for 30 min. The samples were then purified using MinElute PCR purification Kit (Qiagen 28004). Library fragments were amplified in 1× NEBNext High Fidelity PCR Master Mix (New England Biolabs M0541) and 1.25 μM of custom Nextera PCR primers 1 and 2. Number of amplification cycles for each sample was determined by qPCR to prevent saturation. Final libraries were once again purified using MinElute PCR purification kit (Qiagen 28004) and sequenced on Illumina Hiseq4000 to generate paired-end 50 bp reads.

Paired-end reads were trimmed for Illumina adaptor sequences and transposase sequences using a customized script and mapped to mm10 using Bowtie2 (2.3.4.1)[98] with parameters -p 4–very-sensitive. Duplicate reads were removed with Samtools[94]. Narrow peaks were called using MACS2[99] with parameters–nomodel–extsize 73–shift -37 -q 0.001. Peaks from all samples were merged into a unique peak list, and raw read counts mapped to each peak [using bedtools multicov (Quinlan laboratory, University of Utah, Salt Lake City, UT) for each individual sample were quantified.

Differentially accessible peaks from the merged union peak list were selected with the DESeq2 R package from Bioconductor. Cutoffs were set at log₂ fold change > 1 or <−1 and adjusted p-value < 0.001. Motif analysis on peak regions was performed using HOMER function[95] (http://homer.ucsd.edu/homer/motif/) "findMotifsGenome.pl" with default parameters to calculate the occurrence of a TF motif in peak regions compared to that in background regions. We used $-\log_{10}(p$-value) to rank the enrichment level of TF motifs.

### In situ hybridization.

Chromogenic In situ hybridization (ISH) for mouse Rest (RNAscope® LS 2.5 probe–Mm-Rest, ACDBio #316258) or mouse Chga (RNAscope® LS 2.5 probe–Mm-Chga, ACDBio #447858) was performed on a Leica Bond RX with Bond Polymer Refine Red Detection kit DS9390. For multiplex ISH and E-cadherin immunofluorescence (IF), ISH was detected with Opal650 using the Opal automation IHC kit (Akoya Biosciences, SKU NEL821001KT), followed by E-cadherin (Cell Signaling Technology 3195, 1:400) on the Leica Bond Max with EnVision+ System-HRP Labeled Polymer Anti-Rabbit (DAKO, K4003) and Opal570 followed by DAPI. Whole sections were scanned using the Olympus VS120 and staining was quantified using the ISH detection algorithms in HALO.

### Immunoassay.

Cells were lysed in RIPA buffer (Thermo Fisher Scientific 89900) supplemented with protease and phosphatase inhibitors (Roche cOmplete ULTRA tablets, Mini, EASYpack 05892970001 and PhosSTOP EASYpack 04906845001). Protein concentration was measured using Pierce™ BCA protein assay kit (Thermo Fisher Scientific 23227). Immunoassays were conducted following the manufacturer's protocol using the capillary-based Simple Western™ assay on the Wes™ system (ProteinSimple). Compass software v4.0.0 was used to analyze and visualize the results. The following antibodies were used: ASCL1 (Santa Cruz sc-374104, 1:200), YAP1 (D8H1X)XP® (Cell Signaling Technology 14074S, 1:2000), NOTCH2 (Cell Signaling Technology 5732, 1:2000), HES1 (Cell Signaling

Technology 11988, 1:50), UCHL1 (Sigma-Aldrich HPA005993, 1:5000), and HSP90 (Cell Signaling Technology 4877, 1:5000).

**Statistics and reproducibility**. Statistical significance was assayed with GraphPad Prism software apart from Wilcoxon rank sum test which was carried out in R. The specific tests used are indicated in the figure legends. Data were presented as mean ± standard deviation unless stated otherwise. All experiments were independently replicated at least twice or findings verified using other orthogonal methods. No data were excluded except for 4 cell lines isolated from *TKO;Rest^{fl/fl}* tumors because *Rest* was not properly deleted as deduced from *Rest* mRNA levels. Mice for in vivo studies were randomized where applicable and sample sizes determined by pilot experiments or previous experiments using similar models. Investigators were not blinded to allocation during experiments and outcome assessment.

**Reporting summary**. Further information on research design is available in the Nature Research Reporting Summary linked to this article.

## Data availability

All genomic sequencing datasets generated in this study are available at Gene Expression Omnibus (GEO) under Super Series GSE171919. Single-cell RNA-seq data of the lungs were downloaded from GSE136580. Source data are provided with this paper.

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

## Acknowledgements

We thank Dr. Anton Berns for the *Trp53*^fl mice, Dr. Gail Mandel for the *Rest*^fl mice, Dr. Tushar Desai for the *Yap1*^fl mice, Dr. Thomas Rando for the *Rbpj*^fl mice, Pauline Chu from the Stanford histology facility, and Dr. Monte Winslow, Dr. Youcef Ouadah and all the members of the Sage lab for their help and support throughout this study. Research reported in this publication was supported by the NIH (grants CA217450 and CA231997 to J.S., grants ARO54780 and ARO46786 to A.E.O.), A*STAR Singapore (NSS (Ph.D.) to Y.T.S. and N.Y.L.), Core Funding to CRUK Manchester Institute (grant A27412 to S.M.P. and K.L.S.), Cancer Research UK (CRUK Manchester Institute (grant no. A27412), the CRUK Manchester Centre (grant no. A25254), and Lung Cancer Centre of Excellence (grant no. A25146 to C.D.).

## Author contributions

Y.T.S. and J.S. designed most of the experiments and interpreted the results, with some help from S.Q.H. (immunostaining, tissue culture), G.L.C. (mouse genetics), and J.S.L. (role of REST/REST ChIP-seq library prep); A.P.D. helped analyzed RNA-seq and ChIP-seq data; N.S.A. performed the first analysis of the ATAC-seq data, and N.Y.L. pursued this analysis under the supervision of A.E.O.; S.M.P. and D.M. performed the RNAscope® analyses under the supervision of K.L.S. and C.D.; Y.T.S. and J.S. wrote the manuscript and prepared the figures with contributions from all authors.

## Competing interests

The authors declare no competing interests.
