## [Peer Review File · Nature Communications]

REVIEWER COMMENTS

Reviewer #1 (Remarks to the Author):

Shue et al. characterize the role of Rest in mediating the trans-differentiation from a neuroendocrine to a non-neuroendocrine fate in multiple contexts i.e., SCLC, lung injury and normal lung development. The authors found that Rest represses a neuroendocrine program, which differs from the *Ascl1*-controlled neuroendocrine program in SCLC and the loss of Rest in the SCLC model decreases tumor initiation but allows the tumors to grow larger. The authors show Rest is not required for the differentiation from PNECs to club cells but loss of Rest does lead to an increase in the size of PNEC clusters in the context of lung injury. Furthermore, in the context of normal lung development loss of Rest increases the number of PNECs as well as NEBs. Mechanistically, the authors describe a circuitry amongst Yap, Notch and Rest to regulate the transition from a neuroendocrine state to a non-neuroendocrine state that is conserved between the lung injury model and SCLC.

The study uses multiple well-crafted model systems to study the neuroendocrine to non-neuroendocrine fate transition. While the authors could have comprehensively profiled these models to draw conclusions that are more substantial on the role of Rest in each context, this still establishes a solid foundation for the field on this highly relevant topic. The authors employ integrative epigenomic and transcriptomic analyses, while these need to be refined to support their conclusions, to state *Ascl1* and Rest regulate distinct neuroendocrine transcriptional programs. Overall, the manuscript is well-written and well-designed characterizing an important fate transition that has implications not only in lung biology but especially in the context of SCLC where several recent studies have suggested neuroendocrine to non-neuroendocrine transition important in therapeutics and disease progression (Ireland et al., 2020; Cancer Cell, Oser et al., 2019; etc...). However, the scientific rationale and the logic of the manuscript are not coherently structured as is and need to be addressed. The detailed and additional critiques are as follows.

Major Critiques

1. In Figure 1, the authors aim to better characterize the Notch-active cell population of SCLC using the TKO; *Hes1*GFP/+ model. To this end, they profiled the expression of various lung cell markers at the RNA and protein levels for *Hes1*+ population.
 - a. Figure 1b and 1c present somewhat inconsistent results, where the expression level changes between GFP-high and GFP-low cells, of AT2 markers (*Sftpc*, *Lamp3* and *Slc34a2*) have a log2 fold change > 2 at the bulk level (1b) and no difference at the single-cell level (1c). How do the authors reconcile the difference?
 - b. Figure 1c shows 100% of the GFP-high cells express *Scgb1a1* (CC10) while figure 1d-e shows 25%-50% of GFP-high (*Hes1*+) cells do not express CC10. How do the authors reconcile this difference between single-cell RT-qPCR and immunofluorescence? It would be beneficial to show the proportion of *Hes1*+ (GFP-high) cells in the TKO model for reference, and add the proportion of CC10+ cells in *Hes1*- cells to contrast the difference (if there is).
 - c. The single cell qRT-PCR data suggest that these GFP-high cells have a cluster of cells that express AT1 cell markers. The authors should provide staining with AT1 markers to provide additional support for their claim or restate their claim to include AT1 cells.
 - d. For the results from Figure 1b, GSEA comparing bulk RNA-seq data of GFP^{high}/GFP^{low} and previously published scRNA signatures generated from the normal lung (for example those curated in Travaglini et al., 2020; Nature) would provide stronger support to conclude that these cells are indeed club-cell like rather than relying on the expression of a couple genes.

Is it plausible that the *Hes1*+ cells mainly represent club cell features with smaller proportions of the *Hes1*+ cells having additional features of AT1 cells (but not AT2 cells)? In such case, what would that bi-phenotypic cell population represent from a lung biology perspective or would it be a unique SCLC phenotype? The authors may want to discuss this point more clearly.

2. In Figures 1, 2 and 6 the authors analyze the expression and epigenetic landscape of tumor cells from TKO; *Hes1*GFP/+ mouse. While the original paper (Lim et al., 2017; Nature) describing

this model used TdTomato as a marker for Cre-recombined cells, this manuscript does not describe the use of the marker as a criterion for their FACS selection. The authors should either explicitly state the use of this marker or if it was not used for these analyses the authors may choose to compare the allelic fraction of Rb, Tp53 and/or p130 between normal, tumor cell lines and their samples to determine the purity of their tumors.

The authors should address the same concern for the TKO;Rest model as well. Additionally, the authors should include the proportion of NCAM^{high} ICAM^{low} and NCAM^{low} ICAM^{high} in the TKO and TKO;Rest models. More importantly, how does the population distribution compare to the Hes1+/GFPhigh populations? It would be difficult to interpret all the data as a whole for this manuscript without that information.

3. In Figures 2 and 3, the approach to identify the differences between the Ascl1 and Rest cistromes have some weaknesses that need to be addressed.

a. Before integrating the genes from transcriptomic and epigenomic analyses, the authors should first independently examine the overlap between Rest and Ascl1 bound regions as determined by ChIP-seq. A heatmap of Rest and Ascl1 genome-wide binding could better represent unique and common targets. The authors may choose to describe enriched pathways from the epigenomic data.

b. The approach to define Ascl1 targets by integrating Ascl1 ChIP-seq with RNA-seq data from TKO GFPhigh vs GFPlow and crossing with Rest targets is a somewhat circular argument in terms of plausible involvement of the Notch-Rest axis. To fully delineate the difference between Ascl1 and Rest regulations, the authors may choose to –

i. Integrate transcriptomic information from genes downregulated in the context of Ascl1 depletion or upregulated in the context of Ascl1 overexpression. The authors may want to consider the use of publicly available datasets.

ii. If the authors aim to determine Ascl1 targets that may be downregulated in the context of active Notch signaling, the Venn diagram should include three sets with the Ascl1 ChIP-seq in TKO and RNA-seq of GFPhigh/GFPlow represented as independent datasets.

c. In Figure 3i the authors suggest that the depressed genes upon Rest KO that are not in their Rest targets are also regulated by Rest from Enrichr analysis. This may simply suggest that their approach to define Rest targets are not optimal. Since the authors have comprehensive datasets to define Rest targets, to most accurately define this set of genes, they should use the overlap of genes downregulated in SCLC cells (Figure 2a) with Rest OE and upregulated genes in TKO;Rest (Figure 3g) and the Rest bound regions from ChIP-seq (Figure 2a), with potentially less stringent thresholds to define each to be more inclusive to account for increased reliability by integration.

4. In Figure 3, the authors should provide additional protein/mRNA expression of Rest to confirm the system and to quantify the proportion of Rest expressing tumor cells in the TKO model as well as how those Rest positive proportions in the TKO model differ in Ascl1 expression and Notch activity to better understand the difference in tumor number and size between TKO and TKO;Rest.

5. In Figure 4, the authors may want to describe the distribution of lung cell types in the Ascl1+specific Rest KO mouse. Does the absence of Rest alter the formation/morphology of PNECs?

6. For Figure 4E, the authors provide insufficient evidence to support their claim in the corresponding lines 219-223. The authors would need to examine the expression of CC10/SYP and proliferation markers at multiple time points post-injury to claim that loss of Rest results in a delay in differentiation to club cells and extends the proliferative state of the PNECs. The current evidence seems to be also consistent with a situation where even in the absence of Rest PNECs can trans-differentiate to club cells in the context of lung injury.

7. In Figure 6, the authors' claim that Yap1 promotes the trans-differentiation is only supported by the growth of YAP1-GFP KP1 cells as adherent cells. Although that is a remarkable phenomenon, the authors may want to strengthen their argument with examining the expression of known neuroendocrine markers in their genetically engineered cells.

8. Overall, there are several gaps in the logic of the paper, the authors shift their focus from the role of Rest in SCLC, tissue repair and lung development, while this does indeed provide several different model systems to support their hypothesis; the authors should restructure the figures and the paper such that the scientific rationales are more coherently presented. A graphical abstract may help convey to the readers the role of Rest in NE to non-NE transition in the context of lung injury, SCLC and lung development.

Minor Critiques

1. Figures 1c and 4a may be better presented with dimensionality reduction plots. It would make it easier for the readers to visualize the different cell types or states. The expression of the cell type markers on such a plot would allow for better presentation of the data.
2. Figure 4: The figure legend title claims are not supported by the evidence presented. It should be properly rephrased.
3. Figure 3h: The color scheme does not match 3g.
4. Figure 4a: The figure legend "non-NE" or "NE" should be clearly separated from "Chga-GFP positive" or "negative" for readability.
5. Figure 6b and lines 269 – 271: It would be more accurate to describe differentially more accessible and less accessible rather than opened and closed.
6. Extended Data Figure 1e-f: Should include a title for upregulated or downregulated.
7. Extended Data Figure 3h: Should include statistics.
8. Extended Data Figure 3f: The shapes of the samples are hard to be distinguished. Please make them larger and more distinct.
9. Lines 123: It is unclear to which 2-fold change is relative. Local signal? It should be defined either in the results or methods section.
10. Lines 100-102: It is not appropriate to state that they are a homogenous population of club cell cancers. Additional experiments described above would be needed.

Reviewer #2 (Remarks to the Author):

This study addresses the role of REST and YAP in the control of PNEC fate in SCLC, injury repair and development. Their relationship with ASCL1 and Notch signaling was explored. Bulk RNAseq show that Hes1-GFP high cells show conversion from NE to non-NE. ChIP analysis of REST in Hes1-GFP high cells and overexpression of REST in cultured SCLC revealed REST target genes, and 43/141 of these target overlap with ASCL1 targets. Introduction of Rest mutant into the SCLC TKO model led to slightly larger tumors. Naphthalene injury in *Ascl1creERT2;Rest* mutants lead to larger outgrowth. Inactivation of Rest in development led to significant increase of NEB size. ATAC data between NE and non-NE SCLC also revealed differences in TEAD motifs and Yap and Taz expression is increased in RNAseq data. *Ascl1creERT2;Yap1* mutant showed reduced transdifferentiation into non-NE outgrowth.

This is a mechanistic dissection of the transcriptional mechanism that controls NE vs non-NE cell fate, and the findings should be of interest to the readership of this journal. The comments below are aimed at strengthening the support of conclusions.

1. From bulk RNAseq/qRT, the Hes-GFP-high cells express increased club, AT1 and AT2 markers. From single cell qPCR, some of these cells express AT1 markers such as *Ager* and *Aqp5* which are minimally expressed in club cells. It is unclear why the conclusion was that these non-NEs are uniformly club in characteristic.
2. Figure 3a-c, the effects of loss of Rest on tumor number and size are not significant and should

be concluded so, instead of the current conclusion.

3. Figure 4d, 5d, 7d, statistics should be done based on medium per sample, rather than based on each data point above threshold.

4. If Rest and Yap are not expressed in PNECs, it is unclear how inactivation of either gene under *Ascl1creERT2*, which is only active in PNECs, would lead to phenotypes.

5. Fig.5f, it appears that there is increased signals in the parenchyma. Double staining with epithelial markers such as E-Cadherin will be important to demonstrate that the increase is specific to epithelium as expected.

6. Fig.4c, 7c, difficult to see clusters. Zoom in images will help.

7. Fig.7a, is YAP staining nuclear or cytoplasmic? It is important to distinguish, as the two forms of YAP have distinct functions.

8. It appears that the controls are all +/+. Do heterozygous mutants show a phenotype?

Reviewer #3 (Remarks to the Author):

I enjoyed reading this very interesting manuscript by Shue et al.

My only question really is whether the tumors are solely neuroendocrine cell derived. Since there is no lineage tracing involved this reviewer does not understand the evidence for the Hes1 high non-NE SCLC cells being derived from NE-cells.

The subsequent experiments using *Ascl1-CreERT2* are much cleaner.

Why can the authors not generate tumors using that Cre line or perhaps perform multicolor clonal lineage tracing?

The conclusions would be much stronger. But perhaps the authors previously published on this. In that case they might want to refer to their previous findings more clearly.

In any case the experiments using the injury model seem to support their claims.

Sincerely,

Stijn De Langhe

Point-by-point response to Reviewers:

Reviewer #1 (Remarks to the Author):

Shue et al. characterize the role of Rest in mediating the trans-differentiation from a neuroendocrine to a non-neuroendocrine fate in multiple contexts i.e., SCLC, lung injury and normal lung development. The authors found that Rest represses a neuroendocrine program, which differs from the Ascl1-controlled neuroendocrine program in SCLC and the loss of Rest in the SCLC model decreases tumor initiation but allows the tumors to grow larger. The authors show Rest is not required for the differentiation from PNECs to club cells but loss of Rest does lead to an increase in the size of PNEC clusters in the context of lung injury. Furthermore, in the context of normal lung development loss of Rest increases the number of PNECs as well as NEBs. Mechanistically, the authors describe a circuitry amongst Yap, Notch and Rest to regulate the transition from a neuroendocrine state to a non-neuroendocrine state that is conserved between the lung injury model and SCLC.

The study uses multiple well-crafted model systems to study the neuroendocrine to non-neuroendocrine fate transition. While the authors could have comprehensively profiled these models to draw conclusions that are more substantial on the role of Rest in each context, this still establishes a solid foundation for the field on this highly relevant topic. The authors employ integrative epigenomic and transcriptomic analyses, while these need to be refined to support their conclusions, to state Ascl1 and Rest regulate distinct neuroendocrine transcriptional programs. Overall, the manuscript is well-written and well-designed characterizing an important fate transition that has implications not only in lung biology but especially in the context of SCLC where several recent studies have suggested neuroendocrine to non-neuroendocrine transition important in therapeutics and disease progression (Ireland et al., 2020; Cancer Cell, Oser et al., 2019; etc...). However, the scientific rationale and the logic of the manuscript are not coherently structured as is and need to be addressed. The detailed and additional critiques are as follows.

We thank the Reviewer for their overall positive feedback on our work. We agree with the Reviewer that more comprehensive profiles may help further refine the mechanisms underlying the NE to non-NE transition. We hope that the Reviewer will agree with us that thorough single-cell RNA-seq analyses belong to future studies (especially in light of our in-depth analysis of three completely different *in vivo* systems and several mouse models with 9-10 alleles).

We also tried in the revised version of the manuscript and below to better explain the rationale and logic underlying our work.

Major Critiques

1. In Figure 1, the authors aim to better characterize the Notch-active cell population of SCLC using the TKO; Hes1GFP/+ model. To this end, they profiled the expression of various lung cell markers at the RNA and protein levels for Hes1+ population.

a. Figure 1b and 1c present somewhat inconsistent results, where the expression level changes between GFP-high and GFP-low cells, of AT2 markers (Sftpc, Lamp3 and Slc34a2) have a log₂ fold change > 2 at the bulk level (1b) and no difference at the single-cell level (1c). How do the authors reconcile the difference?

We apologize if this was not clear. The discrepancy comes from the baseline expression levels of these genes: although they showed a range of 2 to 10-fold change in expression, the FPKM

varies widely. Levels of *Scgb1a1* are much higher than the *Sftpc/Lamp3/Slc34a2*. From our single-cell qPCR (which includes a pre-amplification step for target of interests to improve detection limits), we found just a few *Hes1*-positive cells that express relatively high levels of *Lamp2/Slc34a2* and other non-club cell markers (unfortunately, we did not manage to capture any cells expressing *Sftpc*). This is in contrast to *Scgb1a1* which was expressed in all *Hes1*-positive cells in our single-cell qPCR, which in turn is found in very high levels by FPKM in our bulk-RNA-seq.

We added below (Fig. R1) the raw FPKM values from the RNA-seq for the Reviewer.

FPKM	Hes1-pos1	Hes1-neg1	Hes1-pos2	Hes1-neg2	Hes1-pos3	Hes1-neg3	Hes1-pos4	Hes1-neg4
Scgb1a1	96184.6623	1181.9808	94912.9767	742.0699677	99125.286	1308.49759	80885.223	606.69578
Sftpc	64.5809862	3.81477933	64.1090422	1.717408719	9.7483592	6.53646295	268.71015	8.1270218
Lamp3	20.7245467	0.39578907	52.3608741	0.207384718	7.2010268	0.26170819	41.536301	0.2896622
Slc34a2	46.1230766	0.44305442	49.9655881	0.486905783	28.01714	0.3763648	59.482665	0.2092051

Figure R1. FPKM data.

To clarify this point in the revised version, we made sure to state that the non-club cell genes were expressed at low levels in the RNA analysis.

b. Figure 1c shows 100% of the GFP-high cells express *Scgb1a1* (CC10) while figure 1d-e shows 25%-50% of GFP-high (*Hes1*+) cells do not express CC10. How do the authors reconcile this difference between single-cell RT-qPCR and immunofluorescence?

We were aware that these two methods gave different results but attributed these differences to the lack of detection of low abundance proteins by immunostaining. We tried a more sensitive method of immunostaining (using TSA amplification) and found a much better agreement with the single-cell RT-qPCR. Please refer to **revised Figure 1e-f** with the new data. We thank the Reviewer for helping us improve these panels.

It would be beneficial to show the proportion of *Hes1*+ (GFP-high) cells in the TKO model for reference, and add the proportion of CC10+ cells in *Hes1*- cells to contrast the difference (if there is).

We thank again the Reviewer for pointing to these inconsistencies that made the initial version of the manuscript more difficult to read. We previously quantified *HES1*+ cells in the *TKO;Hes1^{GFP/+}* model (Lim *et al.*, 2017) by immunostaining for *HES1* and by FACS for GFP. Based on these data, we estimated a range of 5% to more than 50% of *HES1*+ cells and an average of ~25% (data in human tumors were similar). The tumors quantified in this new study fall in this range. We have added the proportion of CC10+ *HES1*- cells in **revised Figure 1f**.

c. The single cell qRT-PCR data suggest that these GFP-high cells have a cluster of cells that express AT1 cell markers. The authors should provide staining with AT1 markers to provide additional support for their claim or restate their claim to include AT1 cells.

Normal club cells indeed express some AT1/AT2 markers (see **Figure 4a, Suppl. Figure 7b** and as described in the lung single cell atlas by Travaglini *et al.*, 2020, PMID: 33208946) but at much lower levels than bona fide AT1/AT2 cells. Therefore, we do not think that the low levels of AT1/AT2 markers expression (as compared to the more robust club cell markers) mean that the *HES1*+ cells are transiting to the AT1/AT2 cell type as well. As suggested by the Reviewer, we

carried out immunostaining for the AT1 marker *AGER* and found that it is not detected (in contrast to the club cell marker *CC10*). We included these data in a **new Figure 1d**.

d. For the results from Figure 1b, GSEA comparing bulk RNA-seq data of *GFPhigh/GFPlow* and previously published scRNA signatures generated from the normal lung (for example those curated in Travaglini et al., 2020; Nature) would provide stronger support to conclude that these cells are indeed club-cell like rather than relying on the expression of a couple genes.

We thank the Reviewer for this interesting suggestion. We tried GSEA but found that it was not useful in this context for determining the lung epithelial cell type closest to *HES1+* cells from bulk RNA-seq. The main reason is that GSEA picks up small differences/trends that might not be biologically relevant. For example, GSEA picked up an AT2 signal even though AT2 genes are only very lowly detected in our single-cell RT-qPCR. We analyzed a few publicly available datasets but are showing the Angelidis et al. 2019 dataset to the Reviewer below (Fig. R2).

Is it plausible that the *Hes1+* cells mainly represent club cell features with smaller proportions of the *Hes1+* cells having additional features of AT1 cells (but not AT2 cells)? In such case, what would that bi-phenotypic cell population represent from a lung biology perspective or would it be a unique SCLC phenotype? The authors may want to discuss this point more clearly.

The Reviewer is raising another interesting point. Currently, based on the homogeneous and strong expression of club cell markers in *HES1+* cells and the very low and variable expression of AT1/AT2 markers in these cells, we consider this population as homogeneous. We have just begun a new research program using single-cell RNA-seq combined to single-cell ATAC-seq to further investigate the heterogeneity of neuroendocrine and non-neuroendocrine cells in SCLC. Our initial results confirm the relative homogeneity of the *HES1+* cells but it will take many more samples and analyses to generate rigorous answers.

To summarize our answer of point #1 raised by the Reviewer: The main point of our first figure is to highlight the similarities between *HES1+* cells and club cells. *HES1+* SCLC cells express high levels of club cell markers and very low levels of other cell types in the lungs. Normal club cells also express similarly low levels of markers of other cell types in the lungs. Our observations provide a rationale to investigate the neuroendocrine to non-neuroendocrine transition downstream of Notch in SCLC, lung injury, and lung development. To further address the role of Notch signaling in the generation of *HES1+* cells, we now provide new data showing deletion of *Rbpj*, an essential co-activator for Notch signaling, abrogates the generation of these non-NE cells (**new Figure 1g-i** and **new Suppl. Figure 2**). To address the point raised by the Reviewer, we also made sure to state That these *HES1+* cells were a “relatively homogeneous population” and that they were “club cell-like cancer cells” so readers are not misled.

2. In Figures 1, 2 and 6 the authors analyze the expression and epigenetic landscape of tumor cells from TKO; Hes1^{GFP/+} mouse. While the original paper (Lim et al., 2017; Nature) describing this model used TdTomato as a marker for Cre-recombined cells, this manuscript does not describe the use of the marker as a criterion for their FACS selection. The authors should either explicitly state the use of this marker or if it was not used for these analyses the authors may choose to compare the allelic fraction of Rb, Tp53 and/or p130 between normal, tumor cell lines and their samples to determine the purity of their tumors. The authors should address the same concern for the TKO;Rest model as well.

We apologize for the confusion. The Reviewer is referring to ED Figure 1 in the Lim *et al.* paper. Indeed, we had used *TKO;Hes1^{GFP/+};Rosa26^{LSL-tdTomato/+}* mice and the purity of cell sorting was assessed by tdTomato expression (ED Figure 1f in that paper). However, ED Figure 1g in the same paper showed the allelic recombination for *Rb*, *p53*, and *p130* of cells sorted from *TKO;Hes1^{GFP/+}* mice *without* the tdTomato reporter, which is the exact same sorting strategy that we have adopted and included in the current manuscript (**Suppl. Figure 1a**). We have used this same strategy in many experiments in the lab, and we always achieve high purity.

As an additional piece of evidence and to directly address the point raised by the Reviewer, we now include a DNA gel electrophoresis image of the genotyping for samples sorted from *TKO;Rest* mice based on NCAM1 and ICAM1 expression. The **new Suppl. Figure 6e** shows complete recombination of the alleles for the three tumor suppressors.

Additionally, the authors should include the proportion of NCAM^{high} ICAM^{low} and NCAM^{low} ICAM^{high} in the TKO and TKO;Rest models.

When we examined the proportion of NCAM^{high};ICAM^{low} and NCAM^{low};ICAM^{high} cells by FACS, we noticed there was a wider spread for the *TKO;Rest* mutants compared to the *TKO* controls as shown in the Figure below (Fig. R3) for the Reviewer. These analyses reach significance but the spread of the % makes it unclear if there is biological relevance behind these observations. In addition, we kept half of each sample for immunostaining and did not observe any difference in proportion of NE and non-NE populations between *Rest* mutant and wild-type tumors in this other assay (**Suppl. Figure 5d-e**).

A possible explanation could be that the non-NE (NCAM1^{low};ICAM1^{high}) cells from the *Rest* mutant tumors survive the stressful tissue dissociation and FACS process better. But we decided not to include these data in the revised version of the manuscript because we cannot explain them, and the HES1/CC10 immunostaining data are used throughout the manuscript. Still these data highlight the fact that the % of NCAM1^{low};ICAM1^{high} non-NE cells in the *Rest* wild-type *TKO* tumors is very similar to the range of HES1⁺ non-NE cells in *TKO* tumors (**Suppl. Figure 5d**), providing additional evidence that the two populations are similar.

More importantly, how does the population distribution compare to the Hes1+/GFP^{high} populations? It would be difficult to interpret all the data as a whole for this manuscript without that information.

We have included an additional FACS analysis as a **new Suppl. Figure 6a** showing NCAM1/ICAM1 expression in tumor cells sorted from *TKO;Hes1^{GFP/+}* mice with HES1+/GFP^{high} cancer cells expressing low levels of NCAM1 and high levels of ICAM1. This FACS analysis told us that sorting for ICAM and NCAM would be a suitable alternative to the *Hes1^{GFP}* reporter.

To summarize our answer of point #2 raised by the Reviewer: By FACS analysis, HES1+/GFP^{high} cancer cells and NCAM1^{low};ICAM1^{high} cancer cells are very similar populations (**new Suppl. Figure 6a**). The fraction of HES1+/GFP^{high} and NCAM1^{low};ICAM1^{high} cancer cells in tumors are similar. When plated, HES1+/GFP^{high} cancer cells and NCAM1^{low};ICAM1^{high} cancer cells adopt the same morphology (attached to the plate – **Figure 3e**). By RNA-seq, these two populations are extremely similar (**Suppl. Figure 6c-d**). Thus, we are confident that the two populations are comparable.

3. In Figures 2 and 3, the approach to identify the differences between the *Ascl1* and *Rest* cistromes have some weaknesses that need to be addressed.

a. Before integrating the genes from transcriptomic and epigenomic analyses, the authors should first independently examine the overlap between *Rest* and *Ascl1* bound regions as determined by ChIP-seq. A heatmap of *Rest* and *Ascl1* genome-wide binding could better represent unique and common targets. The authors may choose to describe enriched pathways from the epigenomic data.

We thank the Reviewer for the suggestion and have added the heatmap as a **new Suppl. Figure 3g**. Comparison of the REST and ASCL1 ChIP-seq datasets found no overlapping binding sites, which is quite a striking observation. Therefore, we decided to instead compare nearest genes mapped to the binding sites as ASCL1 is known to bind at enhancers and further away from transcription start sites.

b. The approach to define *Ascl1* targets by integrating *Ascl1* ChIP-seq with RNA-seq data from *TKO* GFP^{high} vs GFP^{low} and crossing with *Rest* targets is a somewhat circular argument in terms of plausible involvement of the Notch-*Rest* axis. To fully delineate the difference between *Ascl1* and *Rest* regulations, the authors may choose to i. integrate transcriptomic information from genes downregulated in the context of *Ascl1* depletion or upregulated in the context of *Ascl1* overexpression. The authors may want to consider the use of publicly available datasets, ii. If the authors aim to determine *Ascl1* targets that may be downregulated in the context of active Notch signaling, the Venn diagram should include three sets with the *Ascl1* ChIP-seq in *TKO* and RNA-seq of GFP^{high}/GFP^{low} represented as independent datasets.

As suggested by the Reviewer, we performed *Ascl1* knockdown in mouse SCLC cells using shRNAs and have included the data as a **new Suppl. Figure 4** and **Suppl. Tables 7-8**. However, knocking down ASCL1 was detrimental to the cells (see **Suppl. Figure 4b**). This was expected from several studies, including the work of Drs. Jane Johnson and colleagues showing that ASCL1 is an essential promoter of neuroendocrine SCLC cells (PMID: 27452466) and from the work of Dr. François Guillemot showing that ASCL1 can regulate the cell cycle (PMID: 21536733). This phenotype made it challenging to investigate changes in ASCL1 targets specifically related to neuroendocrine programs. Still, our new analysis in this setting confirms that ASCL1 and REST targets are largely non-overlapping (**new Suppl. Figure 4h**). Therefore, we decided to keep the integration of the ASCL1 ChIP-seq with RNA-seq data from *TKO;Hes1^{GFP/+}* tumors to shortlist ASCL1 targets in our analysis in the main figure (**Figure 3**). We did not include the 3-way Venn diagram as our goal was not to identify ASCL1 targets downregulated by Notch signaling but to compare ASCL1 and REST targets.

We thank the Reviewer for this suggestion, as the genes found after knock-down may represent high-confidence targets useful to us and the field in subsequent analyses.

c. In Figure 3i the authors suggest that the depressed genes upon Rest KO that are not in their Rest targets are also regulated by Rest from Enrichr analysis. This may simply suggest that their approach to define Rest targets are not optimal. Since the authors have comprehensive datasets to define Rest targets, to most accurately define this set of genes, they should use the overlap of genes downregulated in SCLC cells (Figure 2a) with Rest OE and upregulated genes in *TKO;Rest* (Figure 3g) and the Rest bound regions from ChIP-seq (Figure 2a), with potentially less stringent thresholds to define each to be more inclusive to account for increased reliability by integration.

The strategy of intersecting all three datasets (REST ChIP, REST OE, *TKO;Rest* vs. *TKO* RNA-seq) suggested by the Reviewer is also a possible approach to shortlist high-confidence REST targets. Readers of the paper will have access to all these data and will be able to choose how they prefer to filter their list of REST targets. The point of Figure 3i was to indicate that the genes derepressed by loss of REST in *TKO* tumors were indeed REST targets, not to identify these targets or help filter a list of “better” targets. In **Figure 3i**, we show that that a higher proportion of REST targets (76/141, 53.9%) than ASCL1 targets (56/559, 10%) were upregulated with the loss of REST. We clarified the text around Figure 3i.

4. In Figure 3, the authors should provide additional protein/mRNA expression of Rest to confirm the system and to quantify the proportion of Rest expressing tumor cells in the *TKO* model as well as how those Rest positive proportions in the *TKO* model differ in *Ascl1* expression and Notch activity to better understand the difference in tumor number and size between *TKO* and *TKO;Rest*.

We have tried over 10 commercial REST antibodies but were unfortunately unable to reliably detect REST via IHC in mouse samples (e.g., still positive signal in knockout samples). We instead performed *in situ* hybridization for *Rest* and have included the quantification of *Rest*-expressing cancer cells in the *TKO* model as a new **Suppl. Figure 5b**. These data confirm the efficient knockout in *TKO;Rest* tumors.

REST expression is significantly upregulated in the HES1+ population (**Suppl. Figure 1c**) and thus correlates with low ASCL1 activity and high Notch signaling. We have included an additional single-cell RT-qPCR (**revised Suppl. Figure 3a**) demonstrating that cells in which *Rest* is detected are exclusively *Hes1-GFP^{high}*, they express higher levels of Notch pathway members, and they are mostly negative for *Ascl1*. These data support our model of *Rest* as a Notch (NICD) target.

5. In Figure 4, the authors may want to describe the distribution of lung cell types in the *Ascl1*+specific *Rest* KO mouse. Does the absence of *Rest* alter the formation/morphology of PNECs?

We apologize if the experimental set-up was unclear: when we use *Ascl1*^{CreER/+} mice, *Rest* is only deleted in ASCL1⁺ PNECs in adult mice one week before injury, not during development. Other lung epithelial cell types are not directly affected by this approach. Moreover, since PNECs do not express REST normally (until induced to transdifferentiate by injury), deletion of *Rest* does not alter PNEC formation/morphology. To clarify this point, we now provide new immunostaining for the PNEC marker CGRP (**new Suppl. Figure 7c**) and quantification of neuroendocrine bodies (NEBs) in non-injured *Ascl1*^{CreER/+}; *Rest* mutant lungs compared to controls (**Figure 4c,d**)

6. For Figure 4E, the authors provide insufficient evidence to support their claim in the corresponding lines 219-223. The authors would need to examine the expression of CC10/SYP and proliferation markers at multiple time points post-injury to claim that loss of *Rest* results in a delay in differentiation to club cells and extends the proliferative state of the PNECs. The current evidence seems to be also consistent with a situation where even in the absence of *Rest* PNECs can trans-differentiate to club cells in the context of lung injury.

We thank the Reviewer for the suggestion. We performed a time-course study looking at onset of CC10 expression instead of cessation of proliferation as it is more feasible to determine the timeframe of the former. The new data support our model and the result of the time-course study showing the delay in onset of CC10 expression in *Rest* mutants are added as **new Figure 4e,f**.

7. In Figure 6, the authors' claim that *Yap1* promotes the trans-differentiation is only supported by the growth of YAP1-GFP KP1 cells as adherent cells. Although that is a remarkable phenomenon, the authors may want to strengthen their argument with examining the expression of known neuroendocrine markers in their genetically engineered cells.

We agree with the Reviewer and included an immunoassay for canonical non-NE (NOTCH, HES1) and NE (ASCL1, UCHL1) markers as a **new Figure 6h** showing upregulation of non-NE genes and downregulation of NE genes at the protein level, thus confirming our morphological observations.

8. Overall, there are several gaps in the logic of the paper, the authors shift their focus from the role of *Rest* in SCLC, tissue repair and lung development, while this does indeed provide several different model systems to support their hypothesis; the authors should restructure the figures and the paper such that the scientific rationales are more coherently presented. A graphical abstract may help convey to the readers the role of *Rest* in NE to non-NE transition in the context of lung injury, SCLC and lung development.

The way we wrote the paper was to introduce non-NE HES1⁺ SCLC cells as a club-like population (Figure 1) and then draw a parallel between the generation of these cancer cells upon Notch pathway activation in SCLC and similar Notch-driven processes in development and in response to injury. Figures 2 and 3 focus on REST in the cancer setting, while Figures 4 and 5 on REST in injury response and embryonic development. Because we cannot study YAP in the embryonic development setting (YAP1 loss prevents lung development), we only have 2 figures on YAP in cancer and injury response (Figures 5 and 6). One alternative presentation would have been to show all the cancer studies first (for Notch, REST, and YAP), then all the injury studies, and finally all the embryonic studies, but when we tried this way, there were more back and forth in how we presented the logic and the rationale. In the revised version, we made sure that the transitions

were presented logically in the revised manuscript. We also now provide a graphical abstract as a **new Figure 8**.

Minor Critiques

1. Figures 1c and 4a may be better presented with dimensionality reduction plots. It would make it easier for the readers to visualize the different cell types or states. The expression of the cell type markers on such a plot would allow for better presentation of the data.

We thank the Reviewer for the suggestion. We tried other approaches to visualize the data but, in the end, felt that our data were best represented as heatmaps (also keeping the visualization like our previous Lim *et al.* manuscript).

2. Figure 4: The figure legend title claims are not supported by the evidence presented. It should be properly rephrased.

We agree with the Reviewer and have edited the title to better reflect the findings, especially in light of the new data in panels e-f.

3. Figure 3h: The color scheme does not match 3g.

We thank the Reviewer for pointing out the mistake, it has been corrected.

4. Figure 4a: The figure legend “non-NE” or “NE” should be clearly separated from “Chga-GFP positive” or “negative” for readability.

We thank the Reviewer for the suggestion and made the change.

5. Figure 6b and lines 269 – 271: It would be more accurate to describe differentially more accessible and less accessible rather than opened and closed.

We thank the Reviewer for the comment and have changed the phrasing to “more opened” and “more closed”.

6. Extended Data Figure 1e-f: Should include a title for upregulated or downregulated.

We thank the Reviewer for the suggestion and added the titles.

7. Extended Data Figure 3h: Should include statistics.

We thank the Reviewer for the comment: statistics (for the fold-change analysis) are in **Suppl. Table 10**.

8. Extended Data Figure 3f: The shapes of the samples are hard to be distinguished. Please make them larger and more distinct.

We have enlarged the panel and a higher resolution version of the figures is now available.

9. Lines 123: It is unclear to which 2-fold change is relative. Local signal? It should be defined either in the results or methods section.

Fold-change is relative to input, we have clarified this in the Methods.

10. Lines 100-102: It is not appropriate to state that they are a homogenous population of club cell cancers. Additional experiments described above would be needed.

We hope that the additional experiments discussed above and the revised text have sufficiently addressed this issue.

Reviewer #2 (Remarks to the Author):

This study addresses the role of REST and YAP in the control of PNEC fate in SCLC, injury repair and development. Their relationship with ASCL1 and Notch signaling was explored. Bulk RNAseq show that Hes1-GFP high cells show conversion from NE to non-NE. ChIP analysis of REST in Hes1-GFP high cells and overexpression of REST in cultured SCLC revealed REST target genes, and 43/141 of these targets overlap with ASCL1 targets. Introduction of Rest mutant into the SCLC TKO model led to slightly larger tumors. Naphthalene injury in *Ascl1creERT2;Rest* mutants lead to larger outgrowth. Inactivation of Rest in development led to significant increase of NEB size. ATAC data between NE and non-NE SCLC also revealed differences in TEAD motifs and Yap and Taz expression is increased in RNAseq data. *Ascl1creERT2;Yap1* mutant showed reduced transdifferentiation into non-NE outgrowth.

This is a mechanistic dissection of the transcriptional mechanism that controls NE vs non-NE cell fate, and the findings should be of interest to the readership of this journal. The comments below are aimed at strengthening the support of conclusions.

We thank the Reviewer for this positive assessment of our work.

1. From bulk RNAseq/qRT, the Hes-GFP-high cells express increased club, AT1 and AT2 markers. From single cell qPCR, some of these cells express AT1 markers such as *Ager* and *Aqp5* which are minimally expressed in club cells. It is unclear why the conclusion was that these non-NEs are uniformly club in characteristic.

This point was also raised by Reviewer #1. The main point of our first figure is to highlight the similarities between HES1⁺ cells and club cells in the lung epithelium. HES1⁺ SCLC cells express high levels of club cell markers and very low levels of other cell types in the lungs. Normal club cells also express similarly low levels of markers of other cell types in the lungs (see **Figure 4a**, **Suppl. Figure 7b**, and as described in the lung single cell atlas by Travaglini *et al.*, 2020, PMID: 33208946).

From our single-cell qPCR (which includes a pre-amplification step for target of interests to improve detection limits), we found just a few *Hes1*-positive cells that express moderate levels of *Lamp2/Slc34a2* and other non-club cell markers (unfortunately, we did not manage to capture any cells expressing *Sftpc*). This contrasts with *Scgb1a1*, which was highly expressed in all *Hes1*-positive cells in our single-cell qPCR and is found in very high levels by FPKM in our bulk-RNAseq. The raw FPKM values are shown in Figure R1 above. To clarify this point in the revised version, we made sure to state that the non-club cell genes were expressed at low levels in the RNA analysis.

To further reconcile some of the RNA and protein expression data, we also tried a more sensitive method of immunostaining (using TSA amplification) and found a much better agreement with the single-cell RT-qPCR. Please refer to **revised Figure 1e-f** with the new data with HES1 and CC10 expression. As suggested by Reviewer #1, we also carried out immunostaining for the AT1 marker AGER and found that it is not detected in HES1⁺ cells (in contrast to the club cell marker CC10). We included these data in a **new Figure 1d**.

Thus, based on the homogeneous and strong expression of club cell markers in HES1⁺ cells and the very low and variable expression of AT1/AT2 markers in these cells, we consider this population as homogeneous. We have just begun a new research program using single-cell RNA-seq combined to single-cell ATAC-seq to further investigate the heterogeneity of neuroendocrine and non-neuroendocrine cell populations in SCLC. Our initial results confirm the relative homogeneity of the HES1⁺ cells but it will take many more samples and analyses to generate rigorous answers. To address the point raised by the Reviewer, we made sure in the revised

version to state that these HES1+ cells were a “relatively homogeneous population” and that they were “club cell-like cancer cells” so readers are not misled.

2. Figure 3a-c, the effects of loss of *Rest* on tumor number and size are not significant and should be concluded so, instead of the current conclusion.

The Reviewer is probably referring to the magnitude of the effects: while statistically significant, are these observations biologically significant? We did not age cohorts of *Rest* wild-type and knockout mice and do not know if the differences observed would result in a change in survival. Based on the variable time of death of *TKO* mice (death being due to either a large tumor blocking the airway or extensive tumor burden in the lungs or the liver), it is unlikely that the differences observed would change survival curves. Also, we now provide new data to the revised manuscript that deletion of *Rbpj*, an essential co-activator for Notch signaling, abrogates the generation of non-NE cells but does not significantly change tumor number/burden (**new Figure 1g-i** and **new Suppl. Figure 2**). While it is possible that the *Rest* mutant partially reprogrammed non-NE cells may have functions that are different from normal non-NE cells and from tumors with no non-NE cells, we agree with the Reviewer that conclusions should be stated carefully for this part. To address the point raised by the Reviewer, we re-phrased the text at the end of this part of the manuscript, removing any strong conclusion regarding these phenotypes. In the future, we plan to challenge these *Rest* knockout tumors with chemotherapy to determine whether in this context the differences with wild-type tumors become more biologically significant, including in terms of survival.

3. Figure 4d, 5d, 7d, statistics should be done based on medium per sample, rather than based on each data point above threshold.

The Reviewer is correct, this would be the proper way of doing the statistical analysis if all PNECs “responded” to the stimulus. However, work from the Krasnow lab (PMID: 31585080) demonstrated that only ~15% of NEBs (presumably those containing NE stem cells) have the capacity to react to injury. Thus, statistics should be done by considering each NEB as a data point not the medium per sample.

4. If *Rest* and *Yap* are not expressed in PNECs, it is unclear how inactivation of either gene under *Ascl1creERT2*, which is only active in PNECs, would lead to phenotypes.

REST and YAP1 are not normally expressed in PNECs but their expression is induced by injury. Therefore, deletion of *Rest* or *Yap1* before the start of the injury experiment prevents them from being induced later as they would in normal PNECs undergoing expansion and fate transition to club cells following damage to the lung epithelium. This transition from NE to non-NE has been well documented by many investigators, as discussed in the manuscript. Previous single-cell RNA-seq data (**Suppl. Figure 10a**) also suggest that rare PNECs do express *Yap1*, perhaps the PNEC “stem cells” or in response to local injury. We hope the revised text clarifies this point.

5. Fig.5f, it appears that there is increased signals in the parenchyma. Double staining with epithelial markers such as E-Cadherin will be important to demonstrate that the increase is specific to epithelium as expected.

The Reviewer raises a good point. It is very unlikely that non-epithelial cells would start expressing neuroendocrine markers but not completely impossible. To address this point, we performed *in situ* hybridization (ISH) for *Rest* together with immunostaining for E-cadherin and found the two

signals to co-localize (**new Figure 5h**), demonstrating that the increase in *Rest* expression is in epithelial cells.

6. Fig.4c, 7c, difficult to see clusters. Zoom in images will help.

Due to the limit on file size when we uploaded the initial version of the manuscript, we are unable to provide higher quality images. However, the images are at high resolution, and we hope that the current version of the manuscript will make it clearer to the Reviewer. At the time of publication, the highest resolution images will be provided.

7. Fig.7a, is YAP staining nuclear or cytoplasmic? It is important to distinguish, as the two forms of YAP have distinct functions.

We analyzed YAP1 localization and found it mainly cytoplasmic with lower levels in the nucleus. YAP1 localization, as the Reviewer mentioned, is important for its function; the nuclear localization supports a model in which YAP1 can be transcriptionally active in these cells. A larger image with a clearer depiction of this is included in a revised **Suppl. Figure 10b**.

8. It appears that the controls are all +/+. Do heterozygous mutants show a phenotype?

We included *Rest* heterozygous mutants in **Figure 5b** but because we saw no phenotype in terms of numbers of PNECs and NEBs, we subsequently did not analyze heterozygotes for the other *Rest* experiments. We did not include heterozygotes for the *Yap1* injury experiments (in part because we were not even sure that YAP1 loss would have an effect because of compensatory effects of TAZ – which was not the case in this context).

Reviewer #3 (Remarks to the Author):

I enjoyed reading this very interesting manuscript by Shue et al. My only question really is whether the tumors are solely neuroendocrine cell derived. Since there is no lineage tracing involved this reviewer does not understand the evidence for the Hes1 high non-NE SCLC cells being derived from NE-cells.

The subsequent experiments using *Ascl1-CreERT2* are much cleaner. Why can the authors not generate tumors using that Cre line or perhaps perform multicolor clonal lineage tracing? The conclusions would be much stronger. But perhaps the authors previously published on this. In that case they might want to refer to their previous findings more clearly. In any case the experiments using the injury model seem to support their claims.

We thank the Reviewer for their positive view of our work.

A lot of work has been done on the cell type(s) of origin for SCLC using mouse models. It is now clear that, in mice, SCLC can arise from both NE and non-NE cell types (see for example our review in 2020 with Dr. Anton Berns, PMID: 32747478). In the SCLC model we use with Ad-CMV-Cre, most tumors arise from CGRP-negative cells whose identity is still not known (they are not SPC+ or CC10+ cells, see PMID: 30228179). It is formally possible that tumors start from a CGRP-negative NE cell type, but very unlikely based on how rare these cells are and the numbers of tumors we generate. Thus, tumors in *TKO* mice start from a non-NE cell type, then form heterogeneous tumors in which some NE cells can generate non-NE cells. One way in which we further addressed how non-NE cells are generated in the revised manuscript by deleting *Rbpj*, an essential co-activator for Notch signaling, in the *TKO* model. This abrogates the generation of these non-NE cells (**new Figure 1g-i** and new **Suppl. Figure 2**). We also clarified the revised text using this sentence: "In this model, the vast majority of tumors are initiated in non-NE lung epithelial cells".

REVIEWER COMMENTS

Reviewer #1 (Remarks to the Author):

Shue et al. have generated a number of new analyses and performed experiments to address the points raised by the reviewers. Main changes include that a graphical abstract to clearly summarize the findings of the manuscript, additional immunofluorescence experiments to validate their single-cell qPCR experiments, and data from *Ascl1* knockdown in the TKO cell line and transcriptomic data to identify *Ascl1* targets. Together, these data support the authors' original findings that *Rest* mediates trans-differentiation from a neuroendocrine to a non-neuroendocrine fate in multiple contexts i.e., SCLC, lung injury and normal lung development with a circuitry amongst *Rest*, *Notch* and *Yap*. However, there are some remaining concerns in the revised manuscript.

1. While the authors' claims have been strengthened with the additional experiments, some experiments are missing appropriate controls –
 - a. In response to the Major critique 1 (c), the authors provide evidence for the lack of AT1 staining in CC10+ cells; however, *Hes1*/GFP staining is missing to claim that *Hes1*+ cells in the tumor do not have AT1 marker expression.
 - b. In response to the Major critique 2, the authors claim that proportions of the NE and non-NE populations are comparable in the TKO and TKO;*Rest* mouse. Given that the apparent difference in FACS analysis in Figure R3 may be confounded by dissociation efficiency, if the authors claim, only with immunostaining in Suppl. Figure 5d/e, that there is no difference based on non-NE markers (*Hes1* and CC10), it is critical that this is accompanied by NE marker staining.
2. The authors note that there is no overlap between the cistromes of *Rest* and *Ascl1*. Their findings are indeed striking. The authors should note, though, the differences in tissue type for *Ascl1* ChIP (primary mouse tumors) and *Rest* (cultured mSCLC cells), and include a description of the number of replicates, as well.
3. In response to the Major critique 3 (c), the authors argue that the goal of Figure 3i was to identify what regulates those de-repressed genes rather than to identify *Rest* targets. The authors should include a statement as to how they reconcile the differences between the transcriptomic data and the epigenetic data. The difference may be confusing to the readers.
4. A statement is missing from the main text on the discrepancy between single-cell qPCR and bulk RNA may be attributable to low levels of expression.

Reviewer #2 (Remarks to the Author):

The authors have responded to most critiques to satisfaction. To critique 3, "3. Figure 4d, 5d, 7d, statistics should be done based on medium per sample, rather than based on each data point above threshold." The Krasnow data is based on limited number assayed. Regardless of what percentage of NEBs contain stem cells, the statistics in these figure panels should be done based on medium per sample, perhaps displayed side-by-side with the current statistical analysis approach.

REVIEWERS' COMMENTS

Reviewer #1 (Remarks to the Author):

Shue et al. have sufficiently addressed the reviewer's comments. Specifically, the authors have modified and clarified their claims in the text.

Reviewer #2 (Remarks to the Author):

We appreciate the authors' additional response, and all critiques are addressed appropriately.

Point-by-point response to Reviewers.

The reviewers did not raise any additional points.

Reviewer #1 (Remarks to the Author):

Shue et al. have sufficiently addressed the reviewer's comments. Specifically, the authors have modified and clarified their claims in the text.

Reviewer #2 (Remarks to the Author):

We appreciate the authors' additional response, and all critiques are addressed appropriately.